# XAB2 dynamics during DNA damage-dependent transcription inhibition

Lise-Marie Donnio[†], Elena Cerutti[†], Charlene Magnani, Damien Neuillet, Pierre-Olivier Mari, Giuseppina Giglia-Mari*

Institut NeuroMyogène (INMG), CNRS UMR 5310, INSERM U1217, Université Claude Bernard Lyon 1, Lyon, France

**Abstract** Xeroderma Pigmentosum group A-binding protein 2 (XAB2) is a multifunctional protein playing a critical role in distinct cellular processes including transcription, splicing, DNA repair, and messenger RNA export. In this study, we demonstrate that XAB2 is involved specifically and exclusively in Transcription-Coupled Nucleotide Excision Repair (TC-NER) reactions and solely for RNA polymerase 2 (RNAP2)-transcribed genes. Surprisingly, contrary to all the other NER proteins studied so far, XAB2 does not accumulate on the local UV-C damage; on the contrary, it becomes more mobile after damage induction. XAB2 mobility is restored when DNA repair reactions are completed. By scrutinizing from which cellular complex/partner/structure XAB2 is released, we have identified that XAB2 is detached after DNA damage induction from DNA:RNA hybrids, commonly known as R-loops, and from the CSA and XPG proteins. This release contributes to the DNA damage recognition step during TC-NER, as in the absence of XAB2, RNAP2 is blocked longer on UV lesions. Moreover, we also demonstrate that XAB2 has a role in retaining RNAP2 on its substrate without any DNA damage.

*For correspondence:
ambra.mari@univ-lyon1.fr

[†]These authors contributed equally to this work

**Competing interest:** The authors declare that no competing interests exist.

## Editor's evaluation

This manuscript will be of interest for individuals working in genome stability, specifically on the repair of UV damage and nucleotide excision repair (NER). The authors report that the transcription-coupled NER factor XAB2 is mobilized after DNA damage and that XAB2 keeps RNA Pol2 engaged on chromatin. XAB2 mobilization appears to be caused by transcription blockage imposed by the DNA damage.

## Introduction

The DNA molecule in our cells' nucleus forms the instruction manual for proper cellular functioning. Unfortunately, the integrity of our DNA is continuously challenged by a variety of endogenous and exogenous agents (e.g., ultraviolet light [UV], cigarette smoke, environmental pollution, oxidative damage, etc.). These DNA lesions interfere with DNA replication, transcription, and cell cycle progression, leading to mutations and cell death, which may cause cancer, inherited diseases, or aging (*Chatterjee and Walker, 2017*).

To prevent the deleterious consequences of persisting DNA lesions, all organisms are equipped with a network of efficient DNA repair systems. One of these systems is the Nucleotide Excision Repair (NER) which removes helix-distorting DNA adducts caused by UV such as Cyclo-Pyrimidine Dimers (CPDs) and 6-4 Photoproducts (6-4PPs) (*Giglia-Mari et al., 2011*).

In mammals, the different steps of NER require more than 30 different proteins that are recruited sequentially to the DNA damage site, as demonstrated by the different studies of NER proteins kinetics (*Moné et al., 2004*; *Politi et al., 2005*; *Rademakers et al., 2003*; *van den Boom et al., 2004*;

*Zotter et al., 2006*). The first step of NER consists of damage recognition, followed by the opening of the DNA duplex, dual incisions on both sides of the damage, excision of 24–32 oligonucleotides containing the damage and, finally, gap-filling by repair DNA synthesis.

NER is divided into two subpathways depending on DNA lesions position within the genome. The Global Genome repair (GG-NER) will detect and repair lesions throughout the genome, whereas Transcription-Coupled repair (TC-NER) is associated with RNA polymerase 2 (RNAP2) to repair lesions on the transcribed strand of active genes (*Marteijn et al., 2014*).

The NER system has been linked to rare human diseases, classically grouped into three distinct NER-related syndromes. These include the highly cancer-prone disorder Xeroderma Pigmentosum (XP) and the two progeroid diseases: Cockayne Syndrome (CS) and Trichothiodystrophy (TTD). Importantly, CS and TTD patients are not cancer prone but present severe neurological and developmental features (*Hanawalt, 1994*).

Xeroderma Pigmentosum group A (XPA)-binding protein 2 (XAB2) is a highly conserved protein of 100 kDa and consists of 15 tetratricopeptide repeat (TPR) motifs that carry out protein–protein interactions. XAB2 protein was identified as a protein interacting with XPA, a NER factor, using a yeast two-hybrid system (*Nakatsu et al., 2000*). Next, it has been shown that XAB2 interacts also with the TC-NER-specific factors, CSA and CSB, and the elongating form of RNAP2 (*Nakatsu et al., 2000*). XAB2 is also essential for early mouse embryogenesis, as demonstrated by the preimplantation lethality observed in XAB2 knockout mice (*Yonemasu et al., 2005*).

Downregulation of XAB2, using either anti-XAB2 or siRNA, inhibits normal RNA synthesis and the recovery of RNA synthesis after UV irradiation (*Kuraoka et al., 2008*; *Nakatsu et al., 2000*). Furthermore, injection of anti-XAB2 in GG-NER-deficient cells significantly reduces UV-induced Unscheduled DNA Synthesis (UDS) during repair (*Nakatsu et al., 2000*). These results suggest the involvement of XAB2 in transcription and TC-NER.

Further studies have shown that XAB2 is a component of the Prp19/XAB2 complex (Aquarius [AQR], XAB2, Prp19, CCDC16, hISY1, and PPIE) or Prp19/CDC5L-related complex required for pre-mRNA splicing (*Kuraoka et al., 2008*). XAB2, as well as PRP19 and AQR, has been involved in the DNA damage response (DDR) (*Maréchal et al., 2014*; *Onyango et al., 2016*; *Sakasai et al., 2017*). Indeed, XAB2 is essential for homologous recombination (HR) by promoting the end resection step (*Onyango et al., 2016*). PRP19 is a sensor of RPA-ssDNA after DNA damage (*Maréchal et al., 2014*) and AQR contributes to the maintenance of genomic stability via regulation of HR (*Sakasai et al., 2017*).

Interestingly, AQR has a role in removing R-loops, a three-stranded nucleic acids structure composed of a DNA:RNA hybrid and the associated nontemplate single-stranded DNA (*Sollier et al., 2014*). These structures can form during transcription, when an RNA molecule emerging from the transcription machinery hybridizes with its DNA template. They are found abundantly in human gene promoters and terminators where RNA processing occurs (*Wang et al., 2018*).

Despite the knowledge acquired in the last decades on XAB2 and its different cellular roles, little is known about the exact crosstalk and dynamics between its diverse cellular functions, specifically between DNA repair transcription and splicing. In this work, we describe the molecular dynamics of XAB2 within the cell after UV-damage induction and during the TC-NER repair process. We determined in vivo that, in the absence of XAB2, Transcription-Coupled repair reactions are impaired, consequently, restart of transcription after UV damage is abolished. Surprisingly, unlike all the other NER proteins studied so far, the mobility of XAB2 is increased after irradiation, and XAB2 shows no accumulation on local UV lesions. This changing dynamic is not restored until DNA repair is completed. Indeed, in damaged TC-NER-deficient cells, XAB2 remains more mobile. Interestingly, we demonstrate that, after DNA damage induction, XAB2 is not released from the splicing complex but is detached from R-loops, a recently XAB2 identified substrate (*Goulielmaki et al., 2021*). Additionally, we investigate the relation between XAB2 and RNAP2, demonstrating that XAB2 retains RNAP2 on its substrate. Moreover, in the absence of XAB2, RNAP2 interacts strongly and durably with both types of UV lesions (6-4PPs and CPDs), suggesting a role of XAB2 in the DNA damage recognition step of TC-NER.

## Results

### XAB2 is involved in TC-NER process

Two decades ago, Tanaka's research group demonstrated the involvement of XAB2 in the NER pathway (*Kuraoka et al., 2008*; *Nakatsu et al., 2000*). However, the dynamics of XAB2 during the

DNA repair process remained to be elucidated. We aimed to study the shuttling between its different functions when DNA damage is induced. Firstly, we wanted to verify that XAB2 is exclusively involved in TC-NER reactions.

The well-known standard assay used to quantify NER activity is the UDS, which measures replication activity outside the S-phase after ultraviolet light (UV-C) treatment. This technique quantifies the refilling of the single-strand DNA gap by the DNA replicative machinery. When we performed an Unschelduled DNA synthesis (UDS) assay in XAB2-silenced cells, no decreased level of UDS was observed (as well as in mock-treated cells; *Figure 1A* blue and black columns, *Figure 1B* and *Figure 1—figure supplement 1*). As a positive control, when we silenced the excision repair factor XPF, we observed a strong reduction in UDS level (*Figure 1A*, red column, *Figure 1B*, and *Figure 1—figure supplement 1*). This result shows that XAB2 is not involved in the GG-NER subpathway, but does not exclude an involvement of XAB2 in the TC-NER subpathway.

The commonly used assay measuring TC-NER activity is the RNA Recovery Synthesis (RRS). This assay measures the newly transcribed RNA by incorporating a nucleoside analog coupled to a fluorophore. The experiment is conducted at different time points after UV irradiation (0, 3, 16, and 24 hr) in order to quantify the decline in transcriptional activity (3 hr after UV damage) and the restart of transcriptional activity (16–24 hr after UV irradiation). In XAB2-silenced cells, no restart of transcription after UV damage was observed (*Figure 1C*, blue column and *Figure 1—figure supplement 2A*), as well as in siXPF-treated cells due to the inability to repair DNA lesions (*Figure 1C*, red column and *Figure 2—figure supplement 2A*) and in contrast with siMock-treated cells (*Figure 1C*, black column and *Figure 1—figure supplement 2A*). In XAB2-silenced cells and in the absence of DNA damage, we observed a decreased level of nascent RNA synthesis when the nucleoside analog EU was incubated for 1 hr, accordingly to the result of Tanaka's group (*Figure 1—figure supplement 2B*; *Kuraoka et al., 2008*; *Nakatsu et al., 2000*). However, when EU is incubated for 2 hr (time point used for RRS assay) we observed an increase of nascent RNA in XAB2-silenced cells compared to control cells (*Figure 1—figure supplement 2B*). As expected, silencing XPF protein does not affect basal transcription (*Figure 1—figure supplement 2C*). RRS results demonstrated an involvement of XAB2 in the TC-NER subpathway but did not discriminate between a role in the repair reaction per se or in the Restart of Transcription after Repair (*Mourgues et al., 2013*).

In order to discriminate this point, we performed an assay designed previously in our group that precisely measures repair replication during TC-NER: the TCR-UDS assay (*Mourgues et al., 2013*). For this assay, we performed the UDS assay in GG-NER-deficient cells using XPC (Xeroderma Pigmentosum complementation group C) mutant cells (XP4PA-SV). The cells were transfected with specific siRNAs and then locally irradiated with UV-C through a filter. In order to precisely localize DNA-damaged areas, a γH2AX coimmunofluorescence labeling was performed, and repair replication was quantified. In siXPF-treated XPC-negative cells, both the GG-NER and the TC-NER pathways are compromised and, as expected, low TCR-UDS levels were observed compared to siMock-treated cells (*Figure 1D*, red and black columns and *Figure 1—figure supplement 3*). Silencing of XAB2 results in a decrease in TCR-UDS levels (*Figure 1D*, blue column and *Figure 1—figure supplement 3*). This result demonstrates a role of XAB2 in the repair reaction itself, its silencing preventing the DNA synthesis associated with the excision of UV lesions on actively transcribed genes.

Next, we decided to investigate by PLA (Proximity Ligation Assay) whether XAB2 can interact with 6-4PPs and CPDs lesion, helix-distorting DNA adducts caused by UV. We observed a strong interaction between XAB2 and 6-4PPs 1 hr after 10 J/m² irradiation (*Figure 2A, C*). This interaction correlates with the amount of 6-4PPs and, as expected, decreases during repair (*Figure 2B, C*), while XAB2 concentration does not change after irradiation (*Figure 2B, C*). The same result is obtained in PLA experiment between XAB2 and CPDs (*Figure 2—figure supplement 1*). These results demonstrate that XAB2 interacts directly with or is in the proximity of 6-4PP and CPD lesions until their removal.

Recently, we demonstrate that a fully functional NER mechanism is necessary for the repair of ribosomal DNA (rDNA), genes transcribed by the RNA polymerase 1 (RNAP1) (*Daniel et al., 2018*). To investigate the involvement of XAB2 in the repair of ribosomal genes, the level of RNAP1 transcription was measured at different time points after UV irradiation by using a specific ribosomal RNA probe coupled to a fluorophore, as described previously (*Daniel et al., 2018*). This probe recognizes the 5' end of the rDNA transcript, the 47S pre-rRNA (upstream from the first site cleaved rapidly during rRNA processing) (a sketch of the 47S is depicted in *Figure 2—figure supplement 2A*). In siMock-treated

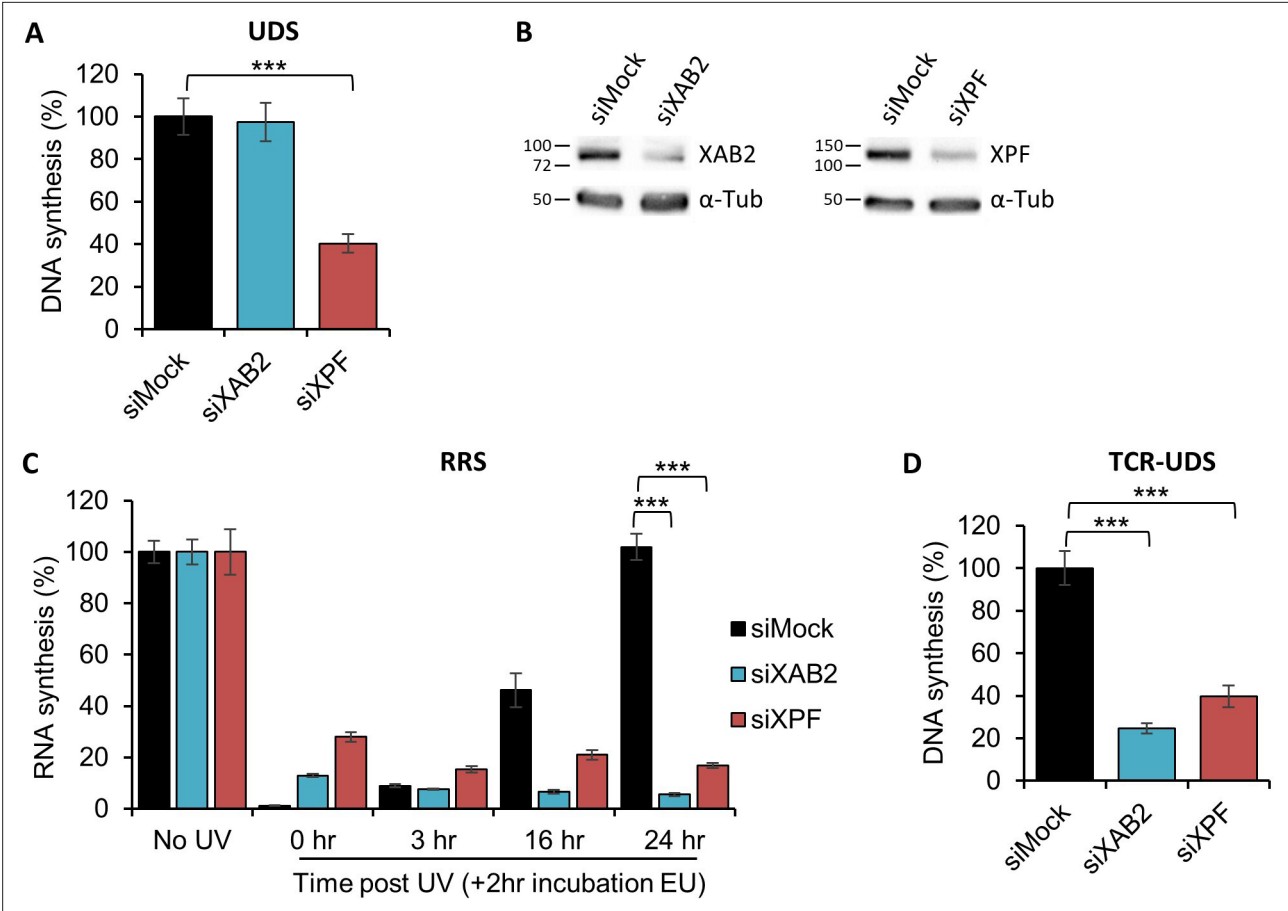

**Figure 1.** XAB2 is involved in DNA repair. (**A**) Quantification of Unscheduled DNA Synthesis (UDS) assay determined by EdU incorporation after local damage (LD) induction with UV-C (100 J/m²) in WT cells (MRC5 cells) treated with siRNAs against indicated factors. Error bars represent the standard error of the mean (SEM) obtained from at least 30 LDs. (**B**) Western blot on whole-cell extracts of MRC5 cells treated with siRNA against indicated factors. (**C**) Quantification of RNA Recovery Synthesis (RRS) assay determined by EU incorporation after UV-C (10 J/m²) exposure in WT cells treated with siRNAs against indicated factors. Error bars represent the SEM obtained from at least 50 cells. (**D**) Quantification of TCR-UDS assay determined by EdU incorporation after LD induction with UV-C (100 J/m²) in GG-NER-deficient cells (XPC−/− cells) treated with siRNAs against indicated factors. Error bars represent the SEM obtained from at least 15 LDs. For all graphs, p-value of Student's test compared to siMock condition: ***<0.001.

The online version of this article includes the following source data and figure supplement(s) for figure 1:

**Source data 1.** Source data for *Figure 1A*: quantification of UDS siXAB2.

**Source data 2.** Source data for *Figure 1C*: quantification of RRS siXAB2.

**Source data 3.** Source data for *Figure 1D*: quantification of TCR-UDS siXAB2.

**Source data 4.** Figures with the uncropped blots and relevant bands clearly labeled for *Figure 1B*: Western blot siXAB2 efficiency.

**Source data 5.** The original files of the full raw unedited gels for *Figure 1B*: Western blot siXAB2 efficiency.

**Figure supplement 1.** UDS siXAB2.

**Figure supplement 2.** RRS siXAB2.

**Figure supplement 2—source data 1.** Source data for *Figure 1—figure supplement 2B, C*: quantification of RNA synthesis.

**Figure supplement 3.** TCR-UDS siXAB2.

cells, we observed a decrease of 47S levels 3 hr after UV-C exposure and the restart of RNAP1 transcription 40 hr after irradiation (*Figure 2—figure supplement 2B*, black column and *Figure 2—figure supplement 2D*). siCSB-treated cells, deficient for TC-NER, presented a low level of rRNA synthesis even 40 hr after UV-C exposure (*Figure 2—figure supplement 2B*, violet column, *Figure 2—figure supplement 2C,D*). In the absence of XAB2, 40 hr after irradiation, the level of 47S returns to the level of non-irradiated condition, supporting a restart of RNAP1 transcription (*Figure 2—figure supplement 2B*, blue column and *Figure 2—figure supplement 2D*).

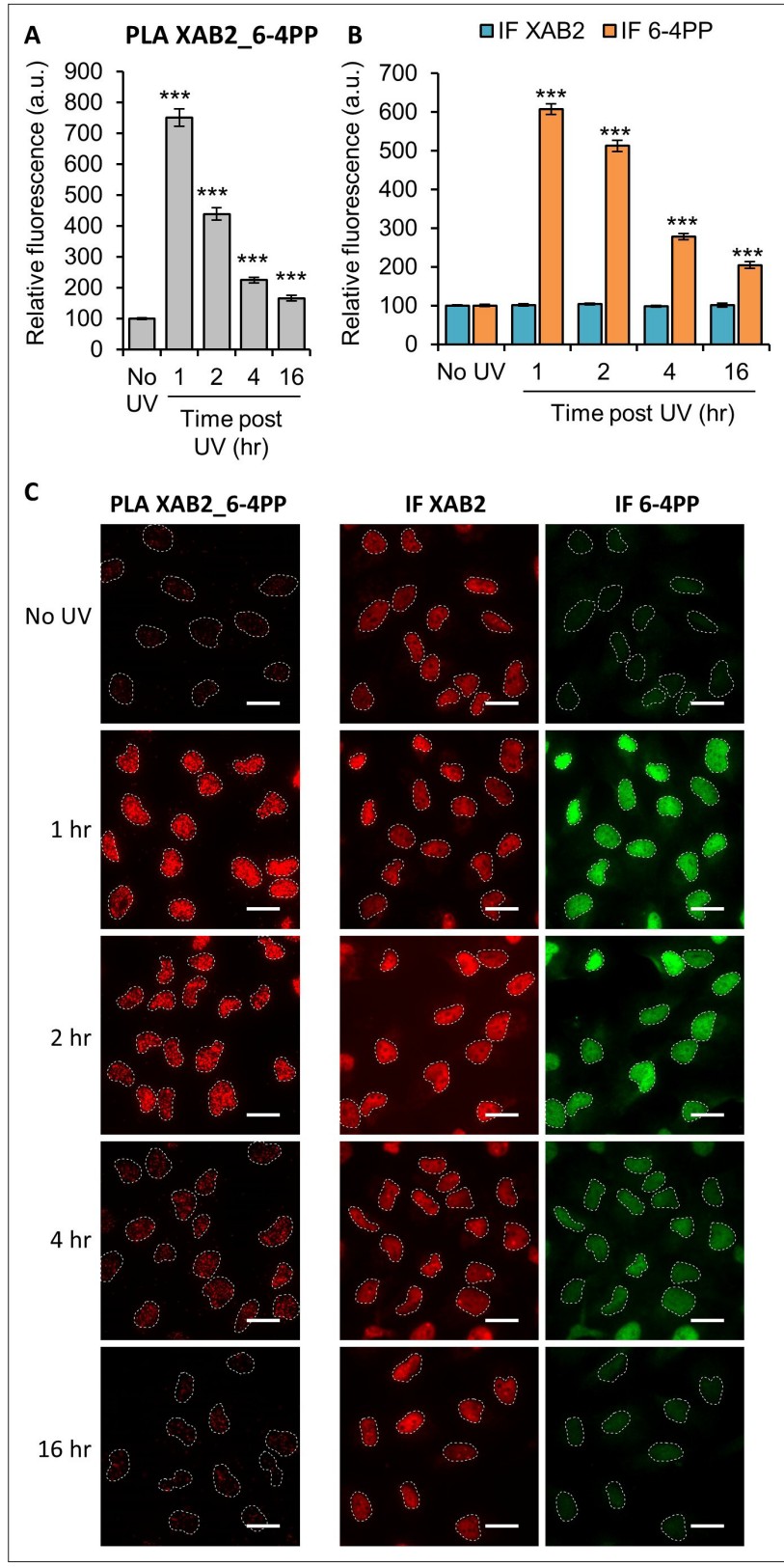

**Figure 2.** XAB2 interacts with the UV lesion 6-4 Photoproduct (6-4PP). Quantification of fluorescent signal in the nucleus against the couple XAB2_6-4PP from Proximity Ligation Assay (PLA) experiment (**A**) or from the immunofluorescence (IF) done in parallel to PLA assay with the same antibodies dilutions (**B**). Error bars represent the standard error of the mean (SEM) obtained from at least 80 cells. P-value of Student's test compared to No UV

*Figure 2 continued*

condition: ***<0.001. (**C**) Representative images of the PLA and IF experiments. Nuclei are delimited by dashed lines. Scale bar: 15 µm.

The online version of this article includes the following source data and figure supplement(s) for figure 2:

**Source data 1.** Source data for *Figure 2A, B*: quantification of PLA and IF XAB2_6-4PP.

**Figure supplement 1.** XAB2 interacts with the ultraviolet light (UV) lesions Cyclo-Pyrimidine Dimer (CPD).

**Figure supplement 1—source data 1.** Source data for *Figure 2—figure supplement 1A, B*: quantification of PLA and IF XAB2_CPD.

**Figure supplement 2.** RNA-FISH siXAB2.

**Figure supplement 2—source data 1.** Source data for *Figure 2—figure supplement 2B*: quantification of RNA-FISH.

**Figure supplement 2—source data 2.** Figures with the uncropped blots and the relevant bands clearly labeled for *Figure 2—figure supplement 2C*: Western blot siCSB efficiency.

**Figure supplement 2—source data 3.** The original files of the full raw unedited gels for *Figure 2—figure supplement 2C*: Western blot siCSB efficiency.

All these results demonstrate that XAB2 has a function in TC-NER repair reactions specifically and exclusively for RNAP2-transcribed genes.

## XAB2-splicing complex is released from the DNA damage area

XAB2 is included in a splicing complex composed of five other proteins: Aquarius (AQR), PRP19, CCDC16, PPIE, and ISY1 (*Kuraoka et al., 2008*). In order to explore how the XAB2-splicing complex behaves after local damage induction, the localization of XAB2, AQR, PRP19, and CCDC16 was revealed by immunofluorescence assays at different time points after local UV irradiation of the cells. In this assay, the fluorescence signal from each protein in the damaged area (visualized by a costaining of γH2AX) was compared to the signal from the rest of the nucleus. Unexpectedly, in contrast with all other NER proteins studied so far, we observed a relatively rapid (1 hr after UV irradiation) release of XAB2, AQR, PRP19, and CCDC16 from the damaged area (*Figure 3A, B*). The proper localization of the XAB2-splicing complex is re-established after the completion of DNA repair reactions when the transcription is fully restarted (16 hr after irradiation; *Figure 3B*).

We thus verified if the entire XAB2-splicing complex was involved in TC-NER or whether only XAB2 played a role in this process. In order to measure the repair capacity of cells silenced for XAB2-related proteins, we performed UDS, TCR-UDS, and RRS experiments in AQR/PRP19/CCDC16/PPIE/ISY1-siRNAs treated cells (*Figure 3—figure supplements 1–3*) and compared the results with XPF-siRNAs treated cell lines. Our results clearly show that none of the cells silenced for XAB2-related proteins are deficient in DNA Repair. Moreover, both GG-NER (*Figure 3—figure supplement 1*) and TC-NER (*Figure 3—figure supplements 2 and 3*) are proficient in the absence of AQR, PRP19, CCDC16, PPIE, or ISY1.

In order to investigate whether XAB2 release from damaged areas was dependent on the TC-NER reaction, the localization of XAB2 was detected and quantified within locally damaged areas in TC-NER-deficient cells: CSA (CS3BE) and CSB (CS1AN) mutant cells. Interestingly, the absence of CSA and CSB did not hinder the release of XAB2 from locally damaged areas. However, this release persisted 16 hr after UV-C exposure (*Figure 3C* blue and red curves compared to black curve and *Figure 3—figure supplement 4*), suggesting that the re-establishment of the proper localization of XAB2 within the nucleus after the DNA repair process depends either on the repair process per se or on the restart of transcription after the achievement of DNA repair reactions.

## XAB2 dynamic during TC-NER

To further analyze XAB2 mobility within the nuclei, we performed SPOT-FRAP (fluorescent recovery after photobleaching) experiments. In this technique, fluorescence molecules are photobleached in a small spot by a high-intensity laser pulse. Subsequently, fluorescence recovery within the bleached area is monitored over time (*Figure 4—figure supplement 1A*). When cells are untreated, the measure of fluorescence recovery corresponds to the protein intrinsic mobility within the living cells (*Figure 4—figure supplement 1A*, black curve). After perturbation of the nuclear environment (e.g.,

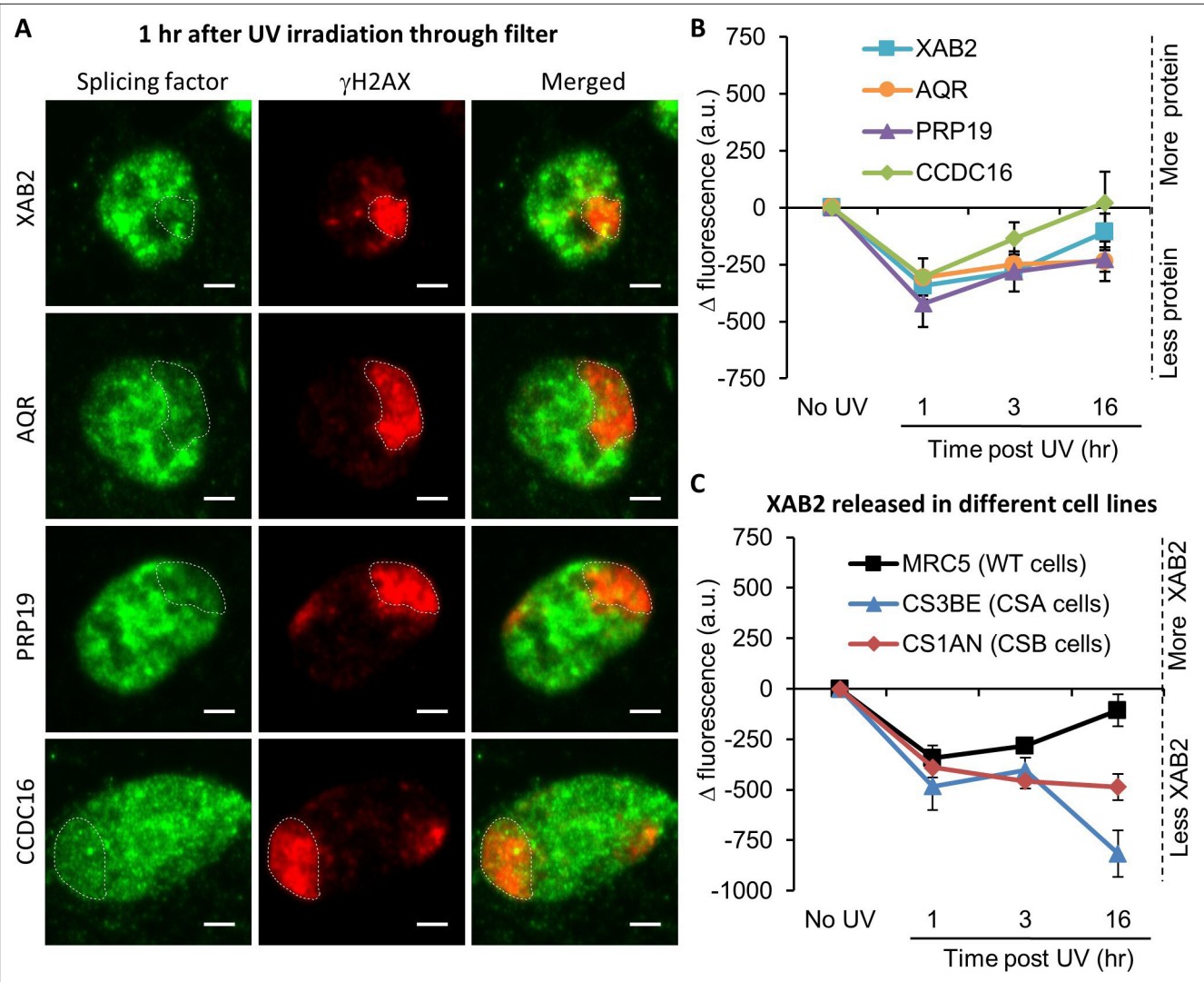

**Figure 3.** Splicing complex is released from DNA damage. (**A**) Representative confocal images of immunofluorescence (IF) against XAB2, AQR, PRP19, or CCDC16 (green) and γH2AX (red) 1 hr after local damage (LD) induction with UV-C (60 J/m²). LDs are indicated by dashed lines. Scale bar: 3 μm. (**B**) Quantification of the IF signal of the different splicing proteins on the LD after different times of recovery. (**C**) Quantification of XAB2 signal on LD in different cell lines after different times of recovery. For both graphs, the signal from the local damage has been subtracted from the background of each cell. Error bars represent the standard error of the mean (SEM) obtained from at least 20 cells.

The online version of this article includes the following source data and figure supplement(s) for figure 3:

**Source data 1.** Source data for *Figure 3B*: quantification of splicing complex IF.

**Source data 2.** Source data for *Figure 3C*: quantification of XAB2 IF.

**Figure supplement 1.** UDS in splicing complex-silenced cell.

**Figure supplement 1—source data 1.** Source data for *Figure 3—figure supplement 1A*: quantification of UDS in splicing complex-silenced cells.

**Figure supplement 1—source data 2.** Figures with the uncropped blots and relevant bands clearly labeled for *Figure 3—figure supplement 1C*: Western blot efficiency of siRNA against splicing complex.

**Figure supplement 1—source data 3.** The orignal files of the full raw unedited gels for for *Figure 3—figure supplement 1C*: Western blot efficiency of siRNA against splicing complex.

**Figure supplement 2.** TCR-UDS in splicing complex-silenced cells.

**Figure supplement 2—source data 1.** Source data for *Figure 3—figure supplement 2A*: quantification of TCR-UDS in splicing complex-silenced cells.

**Figure supplement 3.** RRS in splicing complex-silenced cells.

**Figure supplement 3—source data 1.** Source data for *Figure 3—figure supplement 3A*: quantification of RRS in splicing complex-silenced cells.

**Figure supplement 4.** XAB2 is released from DNA damage also in TC-NER-deficient cells.

DNA damage), a protein can physically interacts with a new substrate or a slower complex, becoming less mobile (*Figure 4—figure supplement 1A*, green curve) or on the contrary can be released from its substrate, becoming more mobile (*Figure 4—figure supplement 1A*, blue curve). Eventually, the protein can also have an unchanged mobility (*Figure 4—figure supplement 1A*, red curve).

We stably transfected a vector expressing a fluorescent version of XAB2 (XAB2-GFP, *Figure 4— figure supplement 1B*) in different SV40-immortalized human fibroblast: wild-type cells (MRC5, *Figure 4—figure supplement 1C*), CSA-deficient cells (CS3BE) and CSB-deficient cells (CS1AN). In order to determine the minimum dose of UV-C needed to detect a significant difference in XAB2 mobility, MRC5 XAB2-GFP cells were irradiated with doses of UV-C ranging from 2 to 16 J/m² and SPOT-FRAP experiments were performed at different time points following UV irradiation (*Figure 4A*). Interestingly, we observed a dose-dependent increase in mobility of XAB2 (*Figure 4A*). Doses of UV-C as weak as 2 and 4 J/m² induced a moderate increase in mobility 3 hr post-irradiation and a recovery of the basal XAB2 mobility within 16 hr post-irradiation (*Figure 4A*, blue and yellow bars). High UV-C doses (16 J/m²) induced a rapid increase in XAB2 mobility (1 hr post-irradiation) and a recovery of the intrinsic mobility 24 hr post-irradiation (*Figure 4A*, red bar). At intermediate doses of 8 J/m² of UV-C, we observed a significant increase in XAB2 mobility during repair (3 hr after UV-C exposure) and the following return to the normal condition once the repair is completed and transcription restarted (16 hr after irradiation) (*Figure 4A*, green bar). Interestingly, in CSA and CSB mutant cells, the increase in XAB2 mobility is also observed after 8 J/m² irradiation and lasted until 24 hr after UV-C exposure (*Figure 4B*), witnessing the fact that in these cells, DNA damage is not repaired and therefore initial intrinsic XAB2 mobility is not restored. Interestingly, without damage, XAB2 mobility is reduced in TC-NER-deficient cells compared to wild-type cells for still unknown reasons (*Figure 4B*, black histogram).

The results of these experiments directed us to explore the possibility that the change in XAB2 mobility was due to transcription inhibition and not really to the repair process itself. In order to verify this hypothesis, XAB2 mobility was measured after DRB (transcription inhibitor) treatment. Surprisingly, results show that XAB2 increased mobility in transcription inhibition conditions is very similar to the one measured upon UV treatment (*Figure 4C*, red curve compared to blue curve).

As for XAB2, the mobility of the late-stage spliceosomes changes after UV irradiation. This mobilization depends on DDR signaling pathways (*Tresini et al., 2015*). Key mediators of DDR are the ATM and ATR kinases, which induce cell cycle arrest and facilitate DNA repair. To demonstrate that the change in XAB2 mobility is due (or not) to the UV-damage response, we realized the same FRAP assays in the presence of ATR and ATM inhibitors. Both drugs did not modify the increase of XAB2 mobility after UV irradiation (*Figure 4C*, green curve and *Figure 4—figure supplement 1D*) demonstrating that variations in XAB2 mobility after DNA damage are triggered and sustained by transcriptional inhibition.

## XAB2 is not released from the splicing complex during DNA repair reactions

The increase of XAB2 mobility after UV-induced transcription inhibition could be explained by either the release of XAB2 from a bigger complex and/or the release from an immobile (or nearly immobile) substrate such as the chromatin or a DNA-related substrate.

In order to distinguish between these two possibilities, we firstly investigated whether, after DNA damage induction, XAB2 dissociates from its splicing partner AQR by immunoprecipitating XAB2 and AQR (*Figure 5—figure supplement 1A*). Interestingly, XAB2 was immunoprecipitated more strongly and consistently 1 hr post-irradiation, time that corresponds to the XAB2 mobility increase. At the same time point, more AQR is also immunoprecipitated. No clear release of XAB2 from AQR was observed at different time points.

In parallel, we also verified by PLA whether the binding of XAB2 to AQR was modified after UV-C irradiation (*Figure 5—figure supplement 1B*). Two hours after UV-C exposure, instead of a release of XAB2 from AQR, we measured a more robust interaction (*Figure 5—figure supplement 1B*). However, this stronger interaction could result from increased AQR concentration 1 hr after UV irradiation (*Figure 5—figure supplement 1C*).

In conclusion, immunoprecipitation or PLA experiment showed that XAB2 is not released from the splicing complex during DNA repair reactions.

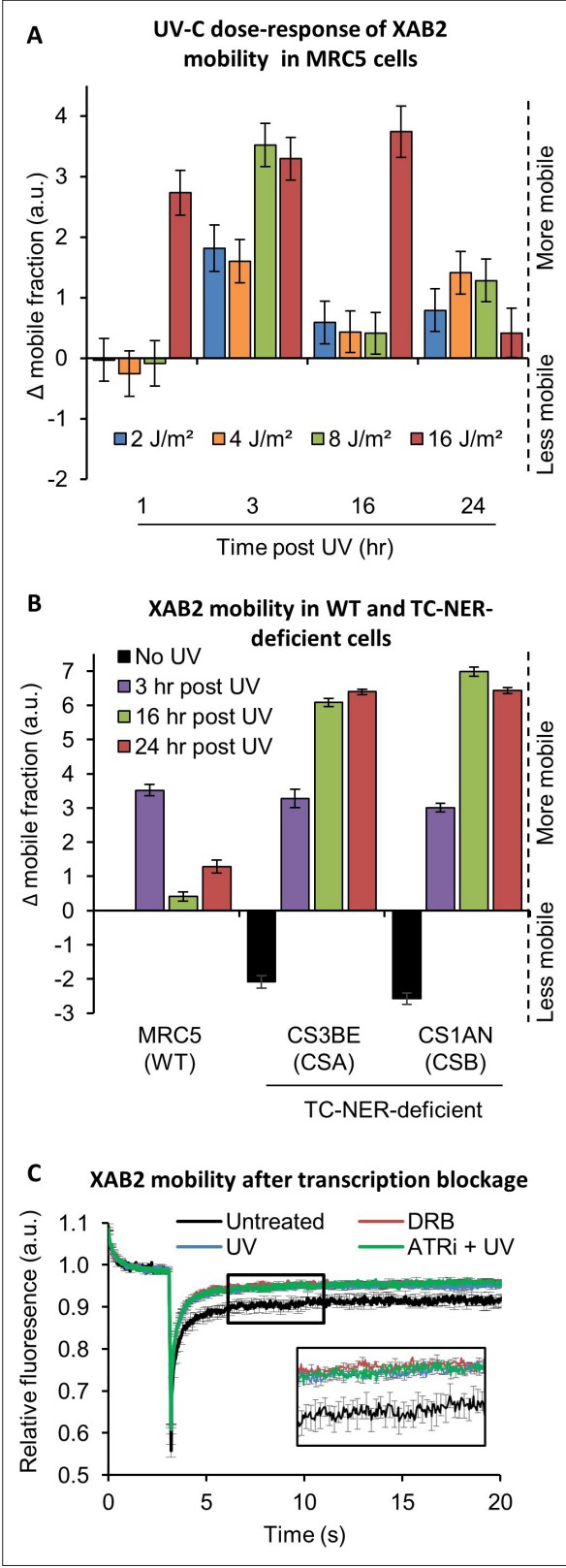

**Figure 4.** XAB2 is dynamic during TC-NER. (**A**) Fluorescent recovery after photobleaching (FRAP) analysis of XAB2-GFP mobility in WT cells. Cells were treated or not with different doses of UV-C (2–16 J/m²) and XAB2 mobility was measured at different time points after UV-C exposure. The No UV condition was used to calculate the change in bound fraction. (**B**) FRAP analysis of XAB2-GFP expressed in WT cells (MRC5-SV) and TC-NER-deficient cells (CSA

*Figure 4 continued on next page*

*Figure 4 continued*

−/− and CSB−/−). Cells were treated or not with 8 J/m² of UV-C. The No UV condition of the WT cell lines was used to calculate the change in bound fraction. (**C**) FRAP analysis of XAB2-GFP mobility in WT cells after treatment with 100 µg/ml of DRB for 2 hr (red line) or with 10 J/m² of UV-C for 3 hr (blue line) or nothing (dark curve). Inhibitor of ATR pathway was added at 10 µM in the medium 1 hr before irradiation (green line). For all graphs, error bars represent the standard error of the mean (SEM) obtained from at least 10 cells.

The online version of this article includes the following source data and figure supplement(s) for figure 4:

**Source data 1.** Source data for *Figure 4A*: FRAP XAB2-GFP with different doses of UV-C.

**Source data 2.** Source data for *Figure 4B*: FRAP XAB2-GFP in different cell lines.

**Source data 3.** Source data for *Figure 4C*: FRAP XAB2-GFP after different treatments.

**Figure supplement 1.** FRAP of XAB2-GFP.

**Figure supplement 1—source data 1.** Source data for *Figure 4—figure supplement 1D*: FRAP XAB2-GFP after different treatments.

## XAB2 is released from R-loops during DNA repair reactions

Interestingly, while trying to immunoprecipitate XAB2 interacting partners during DNA repair reactions, we could observe that systematically and consistently, more XAB2 immunoprecipitated from nuclear extracts 1–3 hr after UV-C irradiation in WT cells (*Figure 5—figure supplement 1A* and *Figure 5A*). At 16 hr post-irradiation, we observed that the amount of XAB2 immunoprecipitated was comparable to the level observed in non-irradiated cells. Moreover, in CSA−/− and CSB−/− cell lines, in which UV lesions on transcribed strands of genes are not repaired, the amount of XAB2 immunoprecipitated remained high all along the time course of the experiment (3 and 16 hr) (*Figure 5A*). These results hinted that the binding between XAB2 and its substrate is not restored in TCR-deficient cells.

AQR has a role in removing DNA:RNA hybrids, commonly known as R-loops structure (*Sollier et al., 2014*) and recently XAB2 was found to be involved in the R-loops resolution (*Goulielmaki et al., 2021*). We thus decided to investigate the possible interaction between XAB2 and R-loops. Immunoprecipitation experiments with the S9.6 antibody show that XAB2 directly interacts with R-loops (*Figure 5B*). To test the specificity of the R-loops antibody, nuclear extracts were treated with Benzonase, an enzyme that specifically degrades DNA:RNA hybrids. Interestingly, after Benzonase treatment, XAB2 is no more immunoprecipitated (*Figure 5B*), suggesting that XAB2 substrate is indeed DNA:RNA hybrids.

The interaction between XAB2 and R-loops was next investigated by performing IF and PLA experiments with the S9.6 antibody. It has been demonstrated that the S9.6 signal is prominently cytoplasmic and nucleolar and derives mainly from ribosomal RNA (*Smolka et al., 2021*). Consequently, to increase the specificity of the S9.6 fluorescent signal, the cytoplasm was removed before fixation (*Figure 5—figure supplement 2A*) and during quantification, the nucleolar signal was subtracted from the nuclear signal (*Figure 5—figure supplement 2B*). Using this analysis method, we observed an increased amount of R-loops in cells silenced for XAB2 or AQR compared to control cells (*Figure 5C* and *Figure 5—figure supplement 3A*).

Subsequently, we examined whether XAB2 is released from R-loops after DNA damage induction by performing a PLA assay. After UV irradiation, we measured a strong and consistent reduction of more than 40% of the interactions between R-loops and XAB2 and between R-loops and AQR (*Figure 5D* and *Figure 5—figure supplement 3B*). These reduced interactions are not caused by a reduction in either R-loops, XAB2, or AQR concentration during DNA repair (*Figure 5E* and *Figure 5—figure supplement 3B*). To verify that this result is specific for R-loops and not coming from a nonspecific interaction of XAB2 with RNAs, we performed the same assays (PLA and IF) in the presence of RNAseH which specifically degrades R-loops structures and not single-stranded RNA (*Figure 5—figure supplement 4A, B*). Our results show that RNAseH reduced the quantity of R-loops (*Figure 5—figure supplement 4C,F*) and in doing so, it also decreased the interaction with both XAB2 (*Figure 5—figure supplement 4C, D*) and AQR (*Figure 5—figure supplement 4E, F*). Next, we verified whether the increase in XAB2 mobility observed after UV irradiation is due to a decreased interaction with messenger RNA (mRNA) (*Figure 5—figure supplement 5*). PLA results show that a reduction in the interaction between XAB2 and mRNA was observed (*Figure 5—figure supplement 5A, C*). However, this reduction paralleled the decrease of mRNA caused by UV-dependent

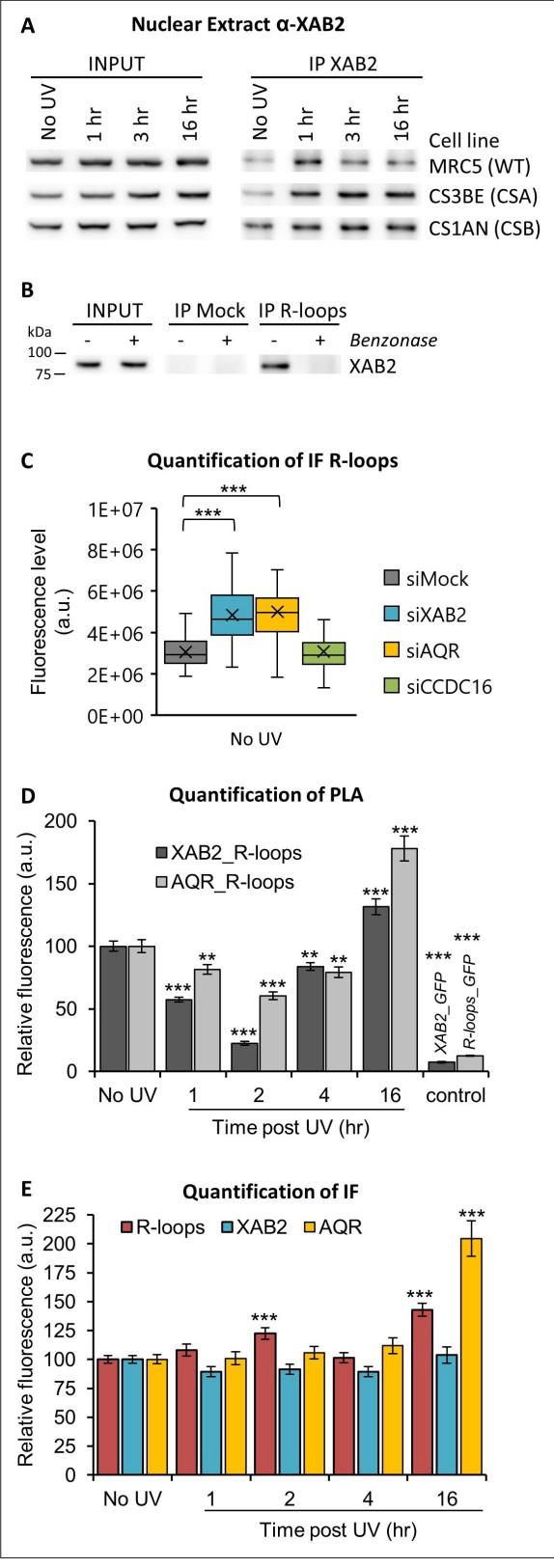

**Figure 5.** XAB2 and AQR are released from R-loops during DNA repair. (**A**) Immunoprecipitation (IF) of XAB2 in nuclear extract from different cell lines treated with 10 J/m² of UV-C at different times. Bound proteins were revealed by Western blotting with antibodies against XAB2. INPUT, 10% of the lysate used for IP reaction. (**B**) IP of R-loops in non-crosslinked chromatin extract from WT cells treated or not with Benzonase. XAB2 bounds to

*Figure 5 continued on next page*

*Figure 5 continued*

R-loops was revealed by Western blotting. INPUT, 10% of the lysate used for IP reaction. (**C**) Quantification of IF against R-loops in WT cells treated with siRNAs against indicated factors. Quantification of fluorescent signal in the nucleus against the couple XAB2_R-loops or AQR_R-loops from PLA experiments (**D**) or from the IF done in parallel to PLA assays (**E**). Error bars represent the standard error of the mean (SEM) obtained from at least 50 cells. P-value of Student's test compared to No UV or siMock condition: **<0.01; ***<0.001.

The online version of this article includes the following source data and figure supplement(s) for figure 5:

**Source data 1.** Source data for *Figure 5C*: quantification of IF R-loops in silenced cells.

**Source data 2.** Source data for *Figure 5D, E*: quantification of PLA and IF XAB2_R-loops and AQR_R-loops.

**Source data 3.** Figures with the uncropped blots and relevant bands clearly labeled for *Figure 5A*: Western blot of IP XAB2.

**Source data 4.** The original files of the full raw unedited gels for *Figure 5A*: Western blot of IP XAB2.

**Source data 5.** Figures with the uncropped blots and relevant bands clearly labeled for *Figure 5B*: Western blot of IP R-loops.

**Source data 6.** The original files of the full raw unedited gels for *Figure 5B*: Western blot of IP R-loops.

**Figure supplement 1.** Interaction of XAB2 with the splicing complex AQR after UV damage.

**Figure supplement 1—source data 1.** Source data for *Figure 5—figure supplement 1B, C*: quantification of PLA and IF XAB2_AQR.

**Figure supplement 1—source data 2.** Figures with the uncropped blots and relevant bands clearly labeled for *Figure 5—figure supplement 1A*: Western blot of IP XAB2 in MRC5.

**Figure supplement 1—source data 3.** The original files of the full raw unedited gels for *Figure 5—figure supplement 1A*: Western blot of IP XAB2 in MRC5.

**Figure supplement 2.** IF R-loops and quantification method.

**Figure supplement 3.** Representatives images of *Figure 5C,D,E*.

**Figure supplement 4.** Specificity of R-loops antibody.

**Figure supplement 4—source data 1.** Source data for *Figure 5—figure supplement 4A*: quantification of PLA and IF R-loops_RNA.

**Figure supplement 4—source data 2.** Source data for *Figure 5—figure supplement 4C, E*: quantification of PLA and IF XAB2_R-loops and AQR_R-loops after treatment with RNAseH.

**Figure supplement 5.** Interaction of XAB2 with RNA.

**Figure supplement 5—source data 1.** Source data for *Figure 5—figure supplement 5A,B*: quantification of PLA and IF XAB2_RNA.

transcription inhibition (*Figure 5—figure supplement 5B, C*), suggesting that the results observed in the PLA XAB2-mRNAs are due mainly to a decrease in mRNAs amount (*Figure 5—figure supplement 5A*).

These results clearly demonstrate that XAB2 is released from R-loops during DNA repair reactions.

## XAB2 is released from CSA and XPG during DNA repair

Because XAB2 has been found to participate specifically in TCR-NER repair reactions (*Figure 1C*), we wanted to investigate whether part of the increased XAB2 mobility observed after UV induction was due to a release from repair complexes. We measured, by PLA, the interactions between XAB2 and CSA, CSB, XPB, or XPG proteins, during TC-NER. Among all the proteins tested, we could observe a clear and consistent release from the CSA protein 2 hr after UV irradiation (*Figure 6A*) and from the XPG protein 1 hr after UV irradiation (*Figure 6C*). The corresponding IF did not show a decreased quantity of CSA or XPG (*Figure 6B, D*) which validated the specificity of the XAB2-CSA and XAB2-XPG decreased interactions at those time points. On the contrary, no clear reduction of interaction was observed between XAB2 and CSB (*Figure 6—figure supplement 1A, B*) or between XAB2 and XPB (*Figure 6—figure supplement 1C, D*).

## XAB2 depletion modifies RNAP2 behavior

Because XAB2 was found to interact with RNAP2 (*Kuraoka et al., 2008*; *Nakatsu et al., 2000*) and because its increased mobility after irradiation depends on the UV transcription inhibition step, we

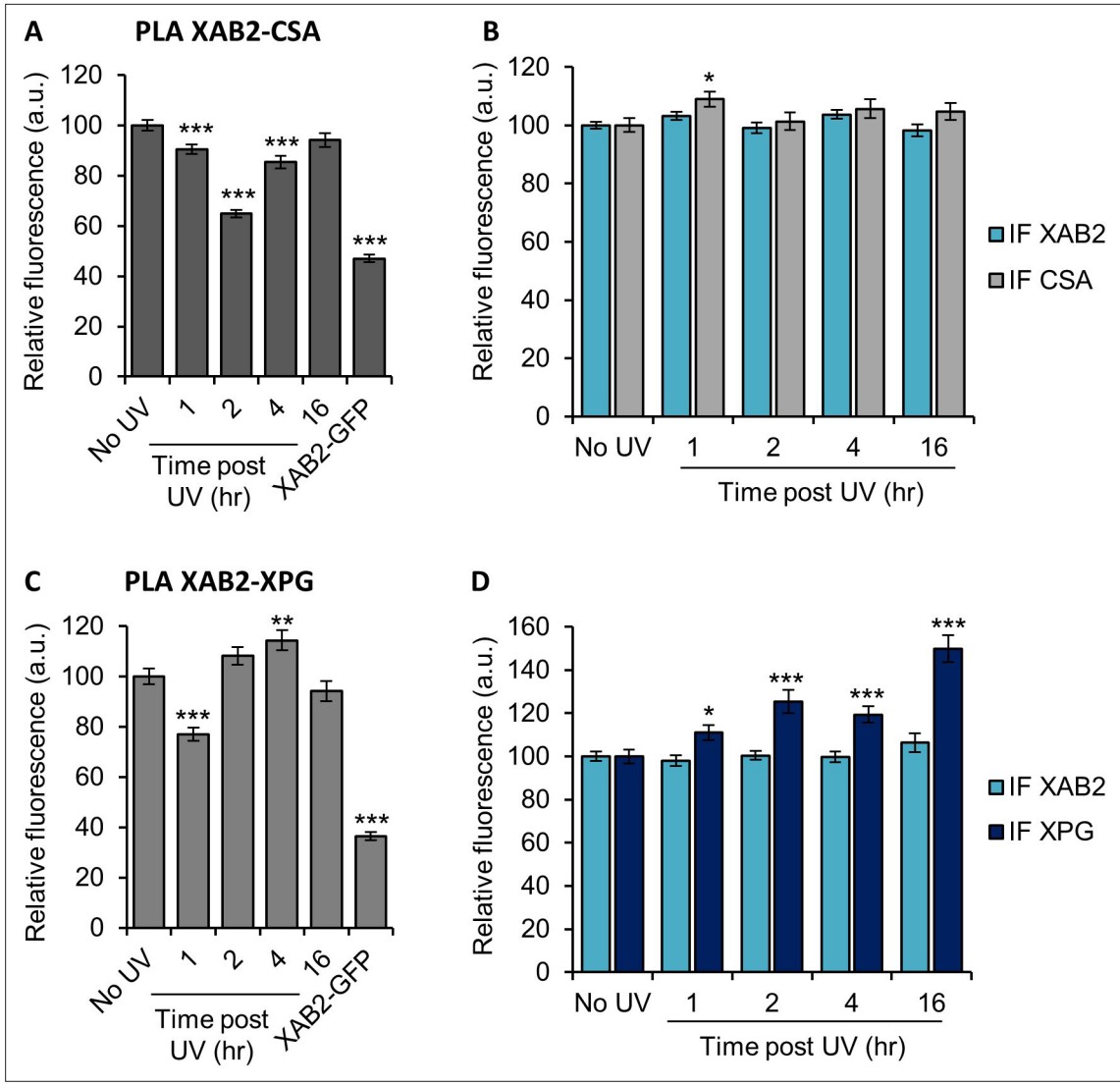

**Figure 6.** XAB2 is released from CSA and XPG during DNA repair. Quantification of fluorescent signal in the nucleus against the couple XAB2-CSA (**A, B**) and XAB2-XPG (**C, D**) from PLA experiment (**A, C**) or from the IF done in parallel to PLA assay (**B, D**). Error bars represent the standard error of the mean (SEM) obtained from at least 80 cells. P-value of Student's test compared to No UV condition: *<0.05; **<0.01; ***<0.001.

The online version of this article includes the following source data and figure supplement(s) for figure 6:

**Source data 1.** Source data for *Figure 6A, B*: quantification of PLA and IF XAB2-CSA.

**Source data 2.** Source data for *Figure 6C, D*: quantification of PLA and IF XAB2-XPG.

**Figure supplement 1.** Interaction of XAB2 with CSB or XPB, repair factors of NER.

**Figure supplement 1—source data 1.** Source data for *Figure 6—figure supplement 1A, B*: quantification of PLA and IF XAB2-CSB.

**Figure supplement 1—source data 2.** Source data for *Figure 6—figure supplement 1C, D*: quantification of PLA and IF XAB2-XPB.

wanted to explore whether XAB2 depletion might influence the overall RNAP2 mobility. In order to investigate this point, we performed FRAP experiments on RNAP2-GFP expressing cells in the presence or in the absence of XAB2 (*Figure 7A*; *Donnio et al., 2019*). Interestingly, depletion of XAB2 significantly increased the mobility of RNAP2 (*Figure 7A*, dark red curve vs. dark blue curve), demonstrating that XAB2 maintains RNAP2 bound to its substrate during transcription. Interestingly, after UV irradiation, RNAP2 mobility did not change significantly (p-value superior at 0.05), both in the presence and absence of XAB2 (*Figure 7A*, light curve vs. dark curve).

Because FRAP experiments might not reveal more subtle changes in protein–substrate interactions, we decided to examine whether the absence of XAB2 might affect the contacts of RNAP2 with

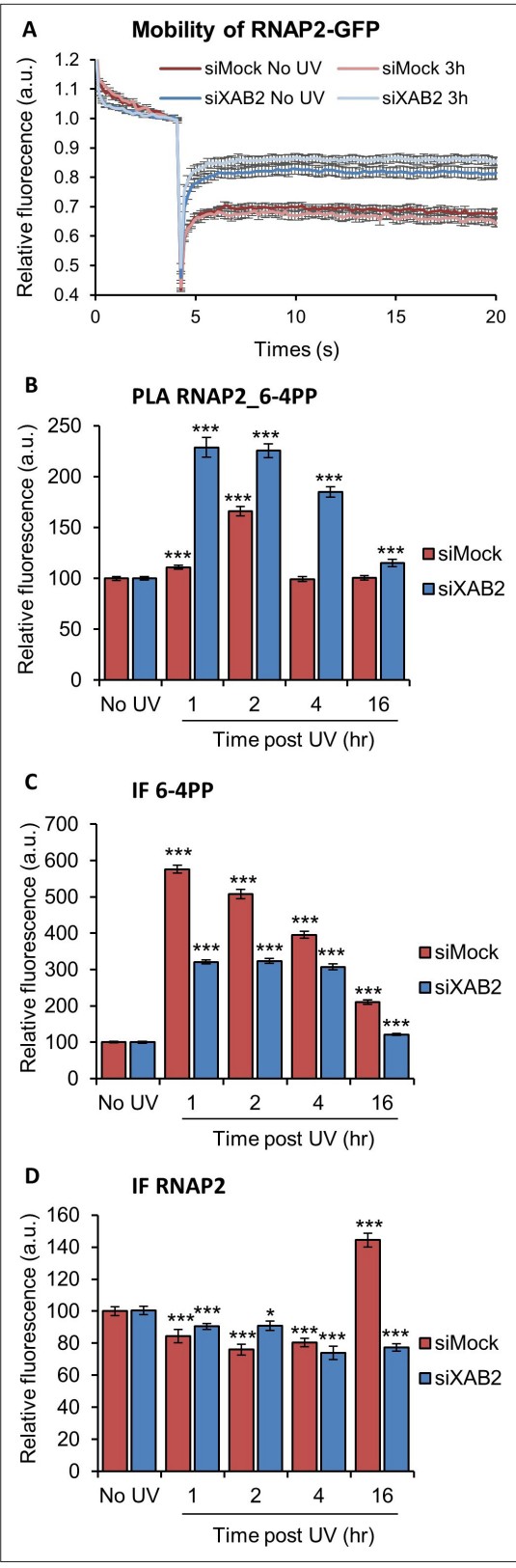

**Figure 7.** RNAP2 behavior is modified without XAB2. (**A**) FRAP analysis of RNAP2-GFP expressing WT cells treated or not with UV-C (10 J/m$^2$) after siRNA-mediated knockdown of the indicated factors. Error bars represent the SEM obtained from at least 10 cells. (**B, C, D**) Quantification of fluorescent signal in the nucleus against the couple RNAP2_6-4PP from PLA experiment (**B**) or from the IFndone in parallel to PLA assay (**C, D**). Error bars represent

*Figure 7 continued on next page*

*Figure 7 continued*

the standard error of the mean (SEM) obtained from at least 80 cells. P-value of Student's test compared to No UV condition: *<0.05; ***<0.001.

The online version of this article includes the following source data and figure supplement(s) for figure 7:

**Source data 1.** Source data for *Figure 7A*: FRAP RNAP2-GFP.

**Source data 2.** Source data for *Figure 7B–D*: quantification of PLA and IF RNAP2_6-4PP.

**Figure supplement 1.** Representatives images of *Figure 7B–D*.

UV lesions using the more sensitive PLA assay. Remarkably, we measured a more robust and persistent interaction of RNAP2 with 6-4PP lesions after UV irradiation in XAB2-silenced cells (*Figure 7B* and *Figure 7—figure supplement 1*). The corresponding IF shows a reduced increase of 6-4PP in XAB2-silenced cells compared to control cells (*Figure 7C* and *Figure 7—figure supplement 1*) while RNAP2 quantity decreases similarly during DNA repair in the presence or absence of XAB2 due to general RNAP2 UV-dependent degradation (*Figure 7D* and *Figure 7—figure supplement 1*). These results demonstrate that, without XAB2, DNA is not properly repaired and as a consequence, RNAP2 interacts longer with UV lesions.

## Discussion

Helix-distorting lesions continuously challenge cell survival by interfering with and blocking fundamental cellular functions, such as transcription and replication. In order to prevent the deleterious effects of these events, cells have developed different mechanisms to restore an undamaged DNA molecule and allow the restart of cellular processes. The importance of rapidly re-establishing perturbed cellular functions is underlined by the presence of a repair mechanism directly coupled with transcription, like the TC-NER.

Two phases can be distinguished during TC-NER events: (1) the actual repair reaction of the damaged strand via the TC-NER subpathway and (2) the resumption of transcription after repair. The experiment known as 'RRS' measures only the restart of transcription after DNA repair completion and thus the involvement of a protein in the general TC-NER process. However, this assay does not discriminate between the two phases of the TC-NER. As a consequence, proteins that are fully proficient in the repair reaction but fail in the transcription restart after DNA repair completion might have the same defect in RRS as those solely deficient in the DNA repair reaction.

Thus far, all studies have demonstrated the involvement of XAB2 in the TC-NER process using only RRS experiments (*Kuraoka et al., 2008*; *Nakatsu et al., 2000*). In this study, we used a specific test developed in-house (*Mourgues et al., 2013*) called TCR-UDS and thus demonstrated that, indeed, XAB2 is solely needed for the repair reaction (*Figure 1D*). This is confirmed by PLA experiments showing an interaction between XAB2 and helix-distorting lesions, 6-4PPs and CPDs, after irradiation (*Figure 2* and *Figure 2—figure supplement 1*). However, these experiments do not clearly discriminate at which step of the repair reaction XAB2 may contribute. Two hypotheses are envisaged: (1) XAB2 functions in damage recognition or (2) in the repair reaction per se.

XAB2 has been found as part of the pre-mRNA splicing complex composed of AQR, PRP19, CCDC16, PPIE, and ISY1 (*Kuraoka et al., 2008*). However, none of these proteins take part in the repair process, underlying the peculiar function of the splicing factor XAB2 in TC-NER repair (*Figure 3—figure supplements 1–3*).

We also demonstrated that, unlike all the other NER proteins studied so far, XAB2 protein is released from the damaged region induced by UV-C exposure (*Figure 3*). Concomitantly, we also observed an increase in XAB2 cellular mobile fraction after UV irradiation (*Figure 4A*). This release from DNA-damaged area and increased mobility is surprising and atypical for a repair protein. However, this behavior has also been observed for late-stage spliceosomes (*Tresini et al., 2015*) and could be explained by the importance of the cell rapidly providing access to the repair machinery.

Surprisingly, the increased XAB2 mobility occurs in the absence of CSA and CSB proteins, with inhibitors of DDR after UV irradiation but also after transcription inhibition (*Figure 4* and *Figure 4—figure supplement 1*). These results strongly suggest that XAB2 remobilization is independent of the repair process but it is more a result of the transcription inhibition induced by the DNA damage.

Moreover, the recovery of intrinsic XAB2 mobility is CSA and CSB dependent probably because in TC-NER defective cells, repair of transcribed genes is deficient and transcription is not recovered. As a consequence, a proper repair and re-establishment of the transcription process are needed to restore XAB2 mobility to normal values.

Previously, we reported that a complete TC-NER mechanism is required to repair the UV lesions present on active rDNA, genes transcribed by RNAP1 (*Daniel et al., 2018*). Notably, both Cockayne syndrome proteins (CSA and CSB) are implicated in this specific repair reaction, as well as the UV-stimulated scaffold protein A (UVSSA), a protein required for the stabilization of CSB specifically after UV irradiation (*Higa et al., 2016*). By measuring the level of ribosomal RNA, we demonstrate that RNAP1 transcription restarts after irradiation in the absence of XAB2 (*Figure 2—figure supplement 2*), meaning that UV lesions present on active rDNA are repaired. As a consequence, XAB2 is involved only in TC-NER of RNAP2-transcribed genes and probably not in the repair of RNAP1-transcribed genes, confirming a likely specific interaction with RNAP2 and reinforcing the idea that RNAP1 and RNAP2 repair processes are distinct although they share common proteins.

FRAP experiments in CSA and CSB mutant cells show a more immobile XAB2 fraction than in WT cells (*Figure 4B*) without any damage induction. Tanaka's group found that XAB2 interacts in vitro with CSA and CSB protein in the absence of DNA damage (*Nakatsu et al., 2000*). In addition, several studies demonstrated the involvement of CSB in transcription regulation (*Boetefuer et al., 2018*). As both CSB and XAB2 are necessary during the transcription process, it is therefore possible that the absence of CSB will modify XAB2 mobility. However, it is not excluded that in CSA and CSB mutant cells, a low level of unrepaired oxidative damage (*de Waard et al., 2004*) might interfere with the proper XAB2 mobility, eventually modifying the amount of R-loops within these cells. Nevertheless, it is difficult to precisely estimate the exact number of R-loops between different cell types, and this hypothesis is difficult to clearly assess.

The UV-induced remobilization of XAB2 is not explained by its release from the splicing complex, as demonstrated by co-immunoprecipitation and PLA experiments, but it is more the result of the release from chromatin-specific structures, in this particular case the R-loops (*Figure 5*). An R-loop is a three-stranded nucleic acid structure composed of a DNA:RNA hybrid and the associated nontemplate single-stranded DNA. This structure arises naturally in organisms from bacteria to humans, and has a multitude of functions in the cell (*Belotserkovskii et al., 2018*). Our results show that R-loops are a substrate for XAB2, and after DNA damage induction, the interaction between XAB2 and R-loops is strongly reduced. This might explain the increased XAB2 mobility during the TC-NER reaction.

It was previously demonstrated that, in AQR-depleted cells, R-loops formation is induced. These R-loops are actively processed into DNA double-strand breaks by XPF and XPG, the NER endonucleases (*Sollier et al., 2014*). Without any damage, we also observed an increased level of cellular R-loops in both AQR- and XAB2-silenced cells (*Figure 5C*), suggesting an involvement of these two proteins in R-loops resolution, as recently also demonstrated by *Goulielmaki et al., 2021*.

Transcription process and R-loops formation are finely interconnected. Indeed, R-loops formation can cause transcription blockage. Transcription blockage due to DNA damage appears to result in R-loops formation (*Mullenders, 2015*; *Steurer and Marteijn, 2017*). Moreover, transcription activity declines after UV irradiation and mRNAs levels are drastically reduced (*Figure 5—figure supplement 5*). PLA assays show that, after UV irradiation, XAB2 and total RNAs interaction is reduced. However, because the total amount of RNAs is diminished after transcription block, we assume that XAB2_RNAs interaction is lessened because of the intrinsic reduced amount of RNAs. Differently from total RNAs, in our study, we observe that R-loops do not decrease in number after UV damage and transcription inhibition (*Figure 5E*). Therefore, the interaction XAB2_R-loops and AQR_R-loops is specifically hindered after UV damage (*Figure 5D*). However, a careful interpretation of these results should be made because a recent paper demonstrates that the S9.6 antibody, specific for DNA:RNA hybrids, can also recognize other nucleic acid structures in immunofluorescence assay (*Smolka et al., 2021*). To be able to confirm our results, many additional controls were performed: (1) removal of the cytoplasm before fixation (*Figure 5—figure supplement 2A*); (2) subtraction of the nucleolar signal from the nuclear signal (*Figure 5—figure supplement 2B*); (3) treatment with RNAseH which degrade specifically R-loops (*Figure 5—figure supplement 4*); and (4) finally interaction of XAB2 with nascent RNA (EU staining) is different from interaction with R-loops (*Figure 5—figure supplement 5*). Although all controls point to a seeming interaction between XAB2 and R-loops or between AQR and R-loops, we

cannot exclude that XAB2 and AQR do not release from DNA:RNA hybrids but from other nucleic acid structures, for example, double-stranded DNA.

Because XAB2 interacts in vitro with CSA and CSB protein (**Nakatsu et al., 2000**) and recent studies have shown an interaction with XPG (**Goulielmaki et al., 2021**), we decided to verify whether these interactions were modified during the DNA repair process and the concomitant transcription inhibition. Our results show a clear and consistent reduction of interaction between XAB2 and CSA (**Figure 6A, B**) and between XAB2 and XPG (**Figure 6C, D**) but not between XAB2 and CSB or between XAB2 and XPB (**Figure 6—figure supplement 1**). These results suggest that XAB2 intervenes in the early steps of TC-NER or at least in the steps that imply the activity of CSB and TFIIH and most probably the early RNAP2 blocking-recognition step. However, the reduction of interactions between XAB2 and CSA or XPG is less striking than the one with R-loops. As suggested in **Goulielmaki et al., 2021**, another working hypothesis would be that XAB2 is released from CSA and XPG proteins associated with R-loops processing.

The physical interaction between XAB2 and RNAP2 has already been established (**Kuraoka et al., 2008**; **Nakatsu et al., 2000**), but the exact relation between XAB2 and RNAP2 has not yet been disclosed. Without DNA damage, we observed a difference in nascent RNA synthesis in XAB2-silenced cells compared to control cells (**Figure 1—figure supplement 2B**). Since XAB2 functions in both transcription and splicing, it is not surprising that the quantity of nascent RNA is altered in the absence of XAB2 (**Goulielmaki et al., 2021**; **Kuraoka et al., 2008**). Moreover, we have clearly demonstrated by FRAP experiments that RNAP2 mobility is severely affected in the absence of XAB2. Namely, in XAB2-depleted cells, RNAP2 is released from its substrate and its mobility is strongly increased (**Figure 7A**). RNAP2 immobile fraction after UV irradiation does not change significantly in the presence or absence of XAB2. However, we could observe that interactions of RNAP2 with the UV lesions (6-4PPs or CPDs)

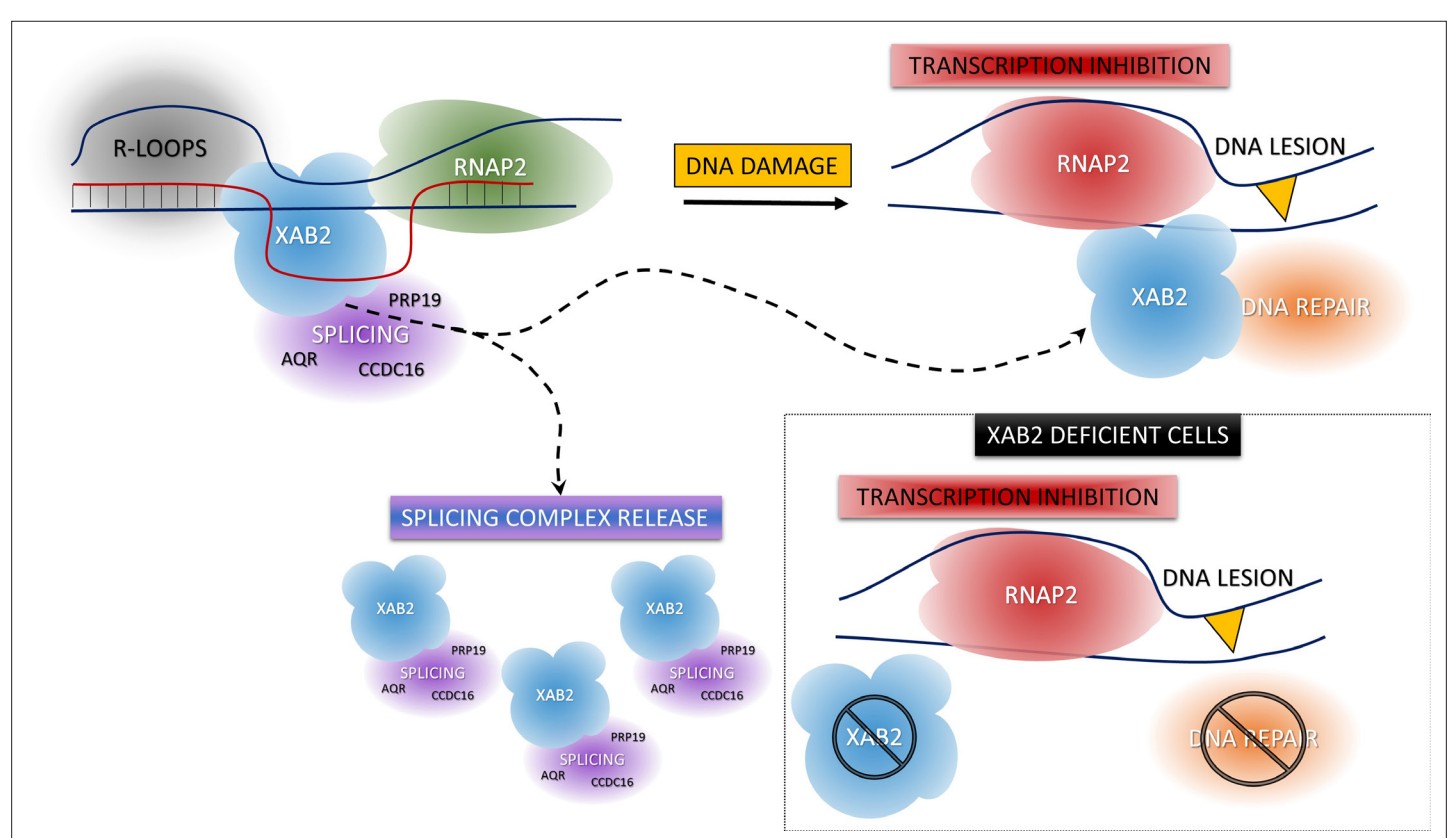

**Figure 8.** Model of XAB2 dynamics during DNA damage-dependent transcription inhibition. Considering our results, a hypothetical model of XAB2 roles and dynamics can be sketched. XAB2 is involved in R-loops removal and pre-mRNA splicing, both processes linked to transcription. After DNA damage induction, transcription is blocked and XAB2 (together with some of the proteins involved in the splicing) is massively released from R-loops allowing a subset of XAB2 molecules to interact with UV-stalled RNA polymerase 2 (RNAP2) and participate in the TC-NER process. In the absence of XAB2, TC-NER is defective and as a consequence, RNAP2 remains longer in the proximity of DNA lesions.

are stronger after DNA damage and last longer in siXAB2-treated cells compared to siMock-treated cells (*Figure 7B–D*). These results suggest that, during transcription, XAB2 helps RNAP2 anchoring to its substrate, whereas during the TC-NER process, it has an effect on stalling of RNAP2 on UV lesions, advocating for a potential role in the DNA damage recognition step.

In conclusion, we describe here an increased mobility of the protein XAB2 during the DNA damage-dependent transcription inhibition. This increased mobility might partly be explained by the release of XAB2 from its substrate R-loops and its partner CSA and XPG. Importantly, we demonstrate that XAB2 plays an anchoring role for RNAP2 to its substrate during transcription and helps RNAP2 to detach from UV lesions after DNA damage. As aconsequence, the absence of XAB2 hinders the overall transcription activity of the cells and severely affects the TC-NER capacity (*Figure 8*).

# Materials and methods

## Cell culture and treatments

The cells used in this study come from Erasmus MC in Rotterdam and were: (1) wild-type SV40-immortalized human fibroblasts (MRC5 [RRID:CVCL_D690]); (2) XPC-deficient SV40-immortalized human fibroblast (XP4PA-SV, GG-NER deficient [RRID:CVCL_6E33]); (3) CSA-deficient SV40-immortalized human fibroblast (CS3BE, TC-NER deficient [RRID:CVCL_F631]); (4) CSB-deficient SV40-immortalized human fibroblast (CS1AN, TC-NER deficient [RRID:CVCL_L472]); (5) MRC5-SV stably expressing XAB2-GFP; (6) CS3BE-SV stably expressing XAB2-GFP; (7) CS1AN-SV stably expressing XAB2-GFP; and (8) MRC5-SV stably expressing RNAP2-GFP. All cell lines are regularly tested negative for mycoplasma contamination.

Immortalized human fibroblasts were cultured in Dulbecco's Modified Eagle Medium(DMEM from Sigma) supplemented with 1% of penicillin and streptomycin (Gibco) and 10% fetal bovine serum (Corning) and incubated at 37°C with 5% $CO_2$.

DNA damage was inflicted by UV-C light (254 nm, 6-W lamp). Cells were globally irradiated with a UV-C dose of 2, 4, 8, 10, or 16 $J/m^2$ or locally irradiated with a UV-C dose of 60 or 100 $J/m^2$ through a filter with holes of 5 µm of diameter (Millipore). After irradiation, cells were incubated at 37°C with 5% $CO_2$ for different periods.

Inhibitor of ATR pathway (VE821) and ATM pathway (KU55933) was added at 10 µM in the medium 1 hr before irradiation.

## Construction and expression of RNAP2-GFP and XAB2-GFP fusion protein

Full-length RNAP2 c-DNA was cloned in-frame into the pEGFP-C1 vector (Clontech), and full-length XAB2 cDNA was cloned in-frame into the pEGFP-N1 vector. Constructs were sequenced prior to transfection.

XAB2-GFP and RNAP2-GFP stably expressing cell lines were produced by transfecting XAB2-GFP or RNAP2-GFP in MRC5, CSA, or CSB cells using FuGENE 6 Transfection Reagent (Promega) according to the manufacturer' protocol. The selection was performed with G418 at 2 mg/ml.

## Transfection of small interfering RNAs

The small interfering RNA (siRNAs) used in this study are: siMock, Horizon, D-001206-14 (10 nM); siXAB2, Horizon, L-004914-01 (20 nM); siXPF, Horizon, M-019946-00 (10 nM); siAQR, CCAGACCA CUUCCCAUUCU (10 nM); siPRP19, GGUGUACAUGGACAUCAAG (10 nM); siCCDC16, GCGAUCUA GUUUCAUUAAA (5 nM); siPPIE, GGCUAUGAGGCAAGUCAAC (5 nM); siISY1, GGAAAUCGAGGU UACAAGU (5 nM) and siCSB, Horizon, L-004888-00 (10 nM). The final concentration used for each siRNA is indicated in parentheses. All siRNA from Horizon are a pool of four different siRNA.

Cells were seeded in 6-well plates and allowed to attach for at least 24 hr. Coverslips were added inside the well if needed for the experiment. Cells were transfected two times with an interval of 24 hr with siRNA using Lipofectamine RNAiMAX reagent (Invitrogen; 13778150) or GenJet (Tebu-Bio), according to the manufacturer's protocol. Experiments were performed between 24 and 72 hr after the second transfection. Protein knockdown was confirmed for each experiment by Western blot.

## RRS assay

MRC5 cells were grown on 18 mm coverslips. siRNA transfections were performed 24 and 48 hr before the RRS assay. RNA detection was done using a Click-iT RNA Alexa Fluor Imaging kit (Invitrogen),

according to the manufacturer's instructions. Briefly, cells were UV-C irradiated (10 J/m²) and incubated for 0, 3, 16, and 24 hr at 37°C. Then, cells were incubated for 2 hr with 100 µM of 5-ethynyl uridine (EU). After fixation with 4% paraformaldehyde (PFA) for 15 min at 37°C and permeabilization with phosphate-buffered saline (PBS) and 0.5% Triton X-100 for 20 min, cells were incubated for 30 min with the Click-iT reaction cocktail containing Alexa Fluor Azide 594. After washing, the coverslips were mounted with Vectashield (Vector). Using ImageJ, the average fluorescence intensity per nucleus was estimated after background subtraction and normalized to not treated cells.

For RRS with siRNA against splicing protein, cells were incubated for 1 hr with 2.5 mM of 5-bromouracile (BrU). Next, the protocol is the same as immunofluorescence. The primary antibody used is mouse anti-BrU diluted in PBS+ (PBS containing 0.15 glycine and 0.5% bovine serum albumin[BSA]) at 1/750 (11170376001 [sigma]).

## RNA fluorescence in situ hybridization

Cells were grown on 18 mm coverslips, washed with warm PBS, and fixed with 4% PFA for 15 min at 37°C. After two washes with PBS, cells were permeabilized with PBS + 0.4% Triton X-100 for 7 min at 4°C. Cells were washed rapidly with PBS before incubation (at least 30 min) with prehybridization buffer: 15% formamide in 2× SSPE (Sodium Chloride-Sodium Phosphate-EDTA) (0.3 M NaCl, 15.7 mM $NaH_2PO_4 \cdot H_2O$, and 2.5 mM EDAT [Ethylenediaminetetraacetic acid] et pH8.0). 35 ng of the probe was diluted in 70 µl of hybridization mix (2× SSPE, 15% formamide, 10% dextran sulfate and 0.5 mg/ml tRNA). Hybridization of the probe was conducted overnight at 37°C in a humidified environment. Subsequently, cells were washed twice for 20 min with prehybridization buffer and once for 20 min with 1× SSPE. After extensive washing with PBS, the coverslips were mounted with Vectashield containing DAPI (Vector). The probe sequence (5′ to 3′) is Cy5-AGACGAGAACGCCTGACACGCACGGCAC.

## UDS or TCR-UDS

MRC5-SV (WT) or XP4PA-SV (GG-NER-deficient) cells were grown on 18 mm coverslips. siRNA transfections were performed 24 and 48 hr before UDS assays. After local irradiation at 100 J/m² with UV-C through a 5 µm pore polycarbonate membrane filter, cells were incubated for 3 or 8 hr (UDS and TCR-UDS, respectively) with 20 µM of EdU (5-ethynyl-2′-deoxyuridine), fixed with 4% PFA for 15 min at 37° C and permeabilized with PBS and 0.5% Triton X-100 for 20 min. Then, cells were blocked with PBS+ (PBS, 0.15% glycine and 0.5% BSA) for 30 min and subsequently incubated for 1 hr at room temperature (RT) with mouse monoclonal anti-γH2AX antibody (Ser139 [Upstate, clone JBW301]) 1:500 diluted in PBS+. After extensive washes with PBS containing 0.5% Triton X-100, cells were incubated for 45 min at RT with secondary antibodies conjugated with Alexa Fluor 594 fluorescent dyes (Molecular Probes, 1:400 dilution in PBS+). Next, cells were washed several times and then incubated for 30 min with the Click-iT reaction cocktail containing Alexa Fluor Azide 488. After washing, the coverslips were mounted with Vectashield containing DAPI (Vector). Images were analyzed as follows using ImageJ and a circle of constant size for all images: (1) the background signal was estimated in the nucleus (avoiding the damage, nucleoli, and other nonspecific signals) and subtracted, (2) the locally damaged area was defined by using the γH2AX staining, and (3) the average fluorescence correlated to the EdU incorporation was then measured and thus an estimation of DNA synthesis after the repair was obtained.

## Immunofluorescence

Cells were plated on 12 or 18 mm coverslips to reach 70% confluence on the day of the staining. After two washes with PBS, cells were fixed with 2% PFA for 15 min at 37°C. Cells were permeabilized by three short washes followed by two washes of 10 min with PBS + 0.1% Triton X-100. Blocking of the nonspecific signal was performed with PBS+ (PBS, 0.5% BSA, 0.15% glycine) for at least 30 min. Then, coverslips were incubated with primary antibody diluted in PBS+ for 2 hr at RT or overnight at 4°C in a moist chamber. After several washes with PBS + 0.1% Triton X-100 (three short washes and two of 10 min) and a short wash with PBS+, cells were incubated for 1 hr at RT in a moist chamber with a secondary antibody coupled to a fluorochrome (Goat anti-mouse Alexa Fluor 488 [A11001, Invitrogen] or 594 [A11005] and Goat anti-rabbit Alexa Fluor 488 [A11008] or 594 [A11012], 1/400 dilution in

PBS+). After the same washing procedure with PBS instead of PBS + 0.1% Triton X-100, coverslips were finally mounted using Vectashield with DAPI (Vector Laboratories).

Treatment with RNAseH (NEB, M0297L) was performed after blocking with PBS+. RNAseH was diluted in PBS+ to put 6 U per coverslip and incubated for 1 hr at 37°C. After washing with PBS+ (one quick and one of 10 min), a primary antibody was added.

For local damage immunofluorescence, the variation of fluorescence in the locally irradiated zone has been calculated, as for the UDS experiment.

For Immunofluorescence with UV lesions antibodies (6-4PP and CPD), a step of DNA denaturation with 0.07 M NaOH freshly diluted in PBS for 5 min at RT was added after permeabilization. After several washes with PBS + 0.1% Triton X-100 (three short washes and two of 10 min), cells were incubated with primary antibody.

For RNA detection, the protocol was adapted from *Petruk et al., 2016*. Briefly, cells were incubated for 2 hr with 100 µM of EU. After fixation, permeabilization, and blocking, a Biotin tag was added thanks to a click-it reaction and then recognized by an antibody.

## Proximity Ligation Assay

PLA experiments were done using Duolink II secondary antibodies and detection kits (Sigma-Aldrich, #DUO92002, #DUO92004, and #DUO92008) according to the manufacturer's instructions. The cells were fixed and permeabilized with the same procedure as immunofluorescence. After blocking 1 hr at 37°C with the Blocking Solution from the kit, the primary antibodies diluted in Antibody Diluent was incubated at 4°C overnight. After one quick and three washes of 5 min with PLA buffer A, Duolink secondary antibodies were added and incubated for 1 hr at 37°C. After the same washing procedure with PLA buffer A, if secondary antibodies were in close proximity (<40 nm), they were ligated together to make a closed circle thanks to the incubation of 30 min at 37°C with the Duolink ligation solution. Then, after the same washing procedure, the DNA is amplified and detected by fluorescence 594 thanks to the incubation of 100 min at 37°C with the Duolink amplification solution. After washing with PLA buffer B, coverslips were mounted using Vectashield with DAPI (Vector Laboratories).

## Cytostripping

To remove the background generated by some antibodies or EdU incorporation, the cytoplasm of the cells was removed before fixation. After two washes with cold PBS, cells were incubated on ice 5 min with cold cytoskeleton buffer (10 mM PIPES [piperazin-N,N'-bis(2-ethanesulfonic acide)] pH 6.8; 100 mM NaCl; 300 mM sucrose; 3 mM $MgCl_2$; 1 mM EGTA [egtazic acid]; 0.5% Triton X-100) followed by 5 min with cold cytostripping buffer (10 mM Tris–HCl pH 7.4; 10 mM NaCl; 3 mM $MgCl_2$; 1% Tween 40; 0.5% sodium deoxycholate). After three gentle washes with cold PBS, cells were fixed.

## Images acquisition and analysis

For RRS, images of the cells were obtained using an Andor spinning disk: Olympus IX 83 inverted microscope, equipped with a Yokaga CSU-X1 Spinning disk Unit and BOREALIS technology for homogeneous illumination. The acquisition software is IQ3.

For RNAFish, UDS, TCR-UDS, and IF of splicing complex after local damage, images of the cells were obtained using a Zeiss LSM 780 NLO confocal laser scanning microscope and the following objective: Plan-Apochromat ×63/1.4 oil DIC (Differential Interference Contrast) M27 or ×40/1.3 oil DIC. The acquisition software is ZEN.

PLA and IF associated with PLA have been performed on a Zeiss Z1 imager right using a ×40/0.75 dry objective. The acquisition software is Metavue.

Images of the cells for each experiment were obtained with the same microscopy system and constant acquisition parameters. All images were analyzed with ImageJ software. All experiments have been performed at least two times and are biological replicates.

Error bars represent the standard error of the mean of the biological replicates. Excel was used for statistical analysis and plotting of all the numerical data. Statistics were performed using a Student's test to compare two different conditions (siMock vs. siRNA X or No UV vs. after irradiation) with the following parameters: two-tailed distribution and two-sample unequal variance (heteroscedastic).

## Primary antibodies used for IF and PLA

Primary antibodies used for immunofluorescence and PLA experiments were anti-6-4PP (mouse, NM-DND-002 [Cosmobio] 1/500 dilution), anti-CPD (mouse, NM-DND-001 [cosmobio], 1/200 dilution), anti-XAB2 (mouse, sc-271037 [Santa Cruz Biotechnology], 1/1000 dilution and rabbit, A303-638A [Béthyl], 1/500 dilution), anti-AQR (IPB160 rabbit, A302-547A [Béthyl], 1/500 dilution), anti-CCDC16 (rabbit, HPA027211 [atlas antibodies], 1/250 dilution), anti-PRP19 (rabbit, ab27699 [abcam], 1/500 dilution), anti-DNA:RNA hybrid clone S9.6 (mouse, MABE1095 [Merck Millipore], 1/100 dilution and rabbit, Ab01137-23.0 [Absolute antibody], 1/100 dilution), anti-Biotin (rabbit, ab1227 [abcam] 1/1000 dilution), anti-CSA (rabbit, GTX100145 [genetex], 1/400 dilution), anti- and anti-CSB (mouse, sc398022 [santa-cruz], 1/200 dilution), anti-XPB (rabbit, sc293 [santa-cruz], 1/500 dilution), and anti-XPG (rabbit, sc84663 [santa-cruz], 1/1000 dilution).

## Fluorescence recovery after photobleaching

FRAP experiments were performed on a Zeiss LSM 780 NLO confocal laser scanning microscope (Zeiss), using a ×40/1.3 oil objective under a controlled environment (37°C, 5% $CO_2$). A narrow region of interest (ROI) centered across the nucleus of a living cell was monitored every 20 ms (1% laser intensity of the 488-nm line of a 25-mW Argon laser) until the fluorescence signal reached a steady state level (after ≈2 s). The same region was then photobleached for 20 ms at 100% laser intensity. Recovery of fluorescence in the bleached ROI was then monitored (1% laser intensity) every 20 ms for about 20 s. Analysis of raw data was performed with the ImageJ software. All FRAP data were normalized to the average prebleached fluorescence after background removal.

XAB2-GFP SPOT FRAP data were analyzed as follows (*Figure 4—figure supplement 1*). The No UV condition's average fluorescence (over all cells) was subtracted from the average fluorescence of the UV-treated conditions. The obtained difference between the two FRAP curves was then summed point by point, starting from the bleach up to the following 100 measurements, that is, the area between the curve of interest and the untreated condition curve.

## Protein extraction

For verification of siRNA efficiency, cells were cultured in a 6-well plate. The coverslip needed for the experiment was displaced before fixation, and cells that remained in the dish were collected. The extraction of total proteins has been performed using the PIERCE RIPA buffer (Thermo, #89900) complemented with PIC (Protease Inhibitor Cocktail from ROCHE).

For immunoprecipitation, cells cultured in 10 cm dishes were harvested by scraping, and the pellet was washed once with PBS supplemented with the PIC. The extraction of nuclear proteins has been performed using the CelLytic NuCLEAR Extraction kit (Sigma-Aldrich) complemented with PIC.

Protein concentration was determined using the Bradford method. The samples were diluted with Laemmli buffer (10% glycerol, 5% β-mercaptoethanol, 3% sodium dodecyl sulfate, 100 mM Tris–HCl [pH 6.8], bromophenol blue) and heated at 95°C before loading on a SDS-PAGE (sodium dodecyl sulfate–polyacrylamide gel electrophoresis).

## Coimmunoprecipitation

For coimmunoprecipitation, 10 µl of protein G magnetic beads (Bio-adembead, Ademtech) were used per IP. 1 µg of anti-XAB2 antibody (rabbit, A303-638A, Bethyl) were bound to the beads in PBS with 3% BSA 3% for 2 hr at 4°C with rotation. 100 µg of nuclear extracts were then incubated with beads–antibodies complex for 2 hr at 4°C with rotation. After two washes at 100 mM salt, two at 150 mM, and one wash at 100 mM, beads were boiled in 2× Laemmli buffer and eluted samples loaded on a SDS–PAGE.

## RNA/DNA hybrid IP

Non-crosslinked MRC5 cells were lysed in 85 mM KCl, 5 mM PIPES (pH 8.0), and 0.5% NP-40 for 10 min on ice. Pelleted nuclei were resuspended in RSB buffer (10 mM Tris–HCl pH 7.5, 200 mM NaCl, 2.5 mM MgCl$_2$) with 0.2% sodium deoxycholate, 0.1% SDS, 0.05% sodium lauroyl sarcosinate, and 0.5% Triton X-100, and extracts were sonicated for 10 min (Diagenode Bioruptor, 60 cycles high power, 10 s ON and 20 s OFF). Extracts were then diluted 1:4 in RSB with 0.5% Triton X-100 (RSB-T)

and subjected to IP with the S9.6 antibody overnight at 4°C. RNaseA was added during IP at 0.1 ng RNaseA per microgram genomic DNA. Then protein G dynabeads (Invitrogen) washed with RSB-T were added and incubated for 3 hr. Beads were washed 4× with RSB-T and 2× with RSB; then eluted in 2× Laemmli buffer for 10 min at 95°C before loading on SDS–PAGE. When indicated, nuclear extracts were treated with 0.25 U/µl Benzonase (Sigma 70664) for 30 min at 37°C before IP.

## Western blot

Proteins were separated on a SDS–PAGE composed of bisacrylamide (37:5:1), blotted onto a PVDF (polyvinylidene difluoride) membrane (0.45 µm Millipore). The membrane was blocked in PBS-T (PBS and 0.1% Tween 20) with 5% milk and then incubated for 2 hr at RT or overnight at 4°C with the following primary antibodies diluted in milk PBS-T: Rabbit anti-XAB2, A303-638A (Bethyl) 1/1000; Mouse anti-XPF, MS-1351-P1 (NeoMarkers) 1/500; Mouse anti-α-Tubulin, T6074 (Sigma-Aldrich) 1/50,000; Rabbit anti-AQR (A302-547A [Bethyl] 1/2000); Rabbit anti-CCDC16 (A301-419A [Bethyl] 1/2000), Rabbit anti-PPIE (ab154865 [abcam] 1/1000); Rabbit anti-PRP19 (ab27692 [abcam] 1/1000); Rabbit anti-ISY1 (ab121523 [abcam] 1/500); Mouse anti-UBF (sc13125 [santa-cruz] 1/500); and Goat anti-CSB (sc10459 [santa-cruz] 1/100).

Subsequently, the membrane was washed repeatedly with PBS-T and incubated 1 hr at RT with the following secondary antibody diluted 1/5000 in milk PBS-T: Goat anti-rabbit IgG HRP conjugate (170-6515; BioRad), Rabbit anti-goat IgG HRP conjugate (172-1034, BioRad) or Goat anti-mouse IgG HRP conjugate (170-6516; BioRad). After the same washing procedure, protein bands were visualized via chemiluminescence (ECL Enhanced Chemiluminescence; Pierce ECL Western Blotting Substrate) using the ChemiDoc MP system (BioRad).

# Acknowledgement

# Additional information

### Funding

| Funder | Grant reference number | Author |
|---|---|---|
| Agence Nationale de la Recherche | ANR-14-CE10-0009 | Giuseppina Giglia-Mari |
| Institut National Du Cancer | PLBIO17-043 | Giuseppina Giglia-Mari |
| Institut National Du Cancer | PLBIO19-126 | Giuseppina Giglia-Mari |
| Ligue Contre le Cancer | 218398 | Giuseppina Giglia-Mari |
| Electricité de France | 218398 | Giuseppina Giglia-Mari |

The funders had no role in study design, data collection, and interpretation, or the decision to submit the work for publication.

### Author contributions

Lise-Marie Donnio, Data curation, Formal analysis, Validation, Investigation, Visualization, Methodology, Writing – original draft, Writing – review and editing; Elena Cerutti, Conceptualization, Formal analysis, Validation, Investigation, Methodology, Writing – original draft; Charlene Magnani, Damien Neuillet, Investigation, Methodology; Pierre-Olivier Mari, Data curation, Software, Supervision, Writing – review and editing; Giuseppina Giglia-Mari, Conceptualization, Supervision, Funding acquisition, Validation, Writing – original draft, Project administration, Writing – review and editing

### Author ORCIDs

Lise-Marie Donnio (iD) http://orcid.org/0000-0002-2414-6034
Elena Cerutti (iD) http://orcid.org/0000-0002-4644-4817
Giuseppina Giglia-Mari (iD) http://orcid.org/0000-0003-2001-1965

Decision letter and Author response

Decision letter https://doi.org/10.7554/eLife.77094.sa1

Author response https://doi.org/10.7554/eLife.77094.sa2

---

## Additional files

### Supplementary files

• MDAR checklist

### Data availability

All data generated or analyzed during this study are included in the manuscript and supporting file.

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
