## [Editor Report]

This manuscript will be of interest for individuals working in genome stability, specifically on the repair of UV damage and nucleotide excision repair (NER). The authors report that the transcription-coupled NER factor XAB2 is mobilized after DNA damage and that XAB2 keeps RNA Pol2 engaged on chromatin. XAB2 mobilization appears to be caused by transcription blockage imposed by the DNA damage.

---

## [Decision Letter]

Thank you for submitting your article "XAB2 Dynamics during DNA Damage-Dependent Transcription Inhibition" for consideration by *eLife*. Your article has been reviewed by 4 peer reviewers, and the evaluation has been overseen by a Reviewing Editor and Kevin Struhl as the Senior Editor.

Essential revisions:

The manuscript is suited for *eLife* and the revision plan as presented by the authors is sound. Here is a list of essential revisions which includes the planned revisions by the authors and some additional points added by the reviewing editor. This implies agreement with the author's choice of revisions NOT to be done but see point #5.

Author list

1) Further PLA experiments with RNAP2 and antibodies against CPD and 6-4PP in the presence and absence of UV and/or siXAB2 will strengthen the point about RNAP2 mobility.

2) Resolve the conflict with Nakatsu et al. 2000 and Kuruoka et al. 2008 regarding the reported increase in transcription with RNAseq experiments with siXAB2 +/- UV. This may resolve the conflict or may not, see #5.

3) Benzonase treatment in IP XAB2 experiments represents a critical control.

4) The plans for the expanded discussion and the changes already implemented are fine.

Editor list

5) Regardless of the outcome of point #2, this conflict will need to be resolved, which may need additional experimentation, if the outcome in inconclusive or cannot explain the discrepancy.

6) I agree that additional experimentation regarding RNAP1 is not needed in the revision, but the conclusion will have to be significantly toned down and limitations be discussed regarding the claimed RNAP2 specificity.

7) The authors should tone down the claim that S9.6 is specific for R-loops, the antibody recognizes also other nucleic acid structures. See J.A. Smolka, L.A. Sanz, S.R. Hartono, F. Chedin Recognition of RNA by the S9.6 antibody creates pervasive artifacts when imaging RNA:DNA hybrids J. Cell Biol., 220 (2021).

[Editors' note: further revisions were suggested prior to acceptance, as described below.]

Thank you for resubmitting your work entitled "XAB2 Dynamics during DNA Damage-Dependent Transcription Inhibition" for further consideration by *eLife*. Your revised article has been evaluated by Kevin Struhl (Senior Editor) and a Reviewing Editor.

The manuscript has been improved but there are some remaining issues that need to be addressed, as outlined below:

The revision addresses the essential points in the previous review. The manuscript requires extensive language editing. There are numerous problems already in the abstract (line 4 'demonstrate', line 7 'becomes', line 8 'change in mobility is restored' needs rephrasing, line 10 'from DNA:RNA hybrids', line 11 'proteins', line 13 'demonstrate').

*Reviewer #4 (Recommendations for the authors):*

The addition of PLA data for the interaction of both RNAP and XAB2 with UV photoproducts is very welcome, and the data quite clearly show increased interaction with damage for both proteins. However, these results actually demand some rethinking of the conclusions.

The authors addressed the essential points for the revision but the manuscript requires extensive language editing.

[Editors' note: this paper was reviewed by Review Commons.]

---

## [Author Response]

Essential revisions:The manuscript is suited for eLife and the revision plan as presented by the authors is sound. Here is a list of essential revisions which includes the planned revisions by the authors and some additional points added by the reviewing editor. This implies agreement with the author's choice of revisions NOT to be done but see point #5.Author list1) Further PLA experiments with RNAP2 and antibodies against CPD and 6-4PP in the presence and absence of UV and/or siXAB2 will strengthen the point about RNAP2 mobility.

This PLA experiment has been performed and we observed more interactions between RNAP2 and UV lesions, 6-4PPs and CPDs, in siXAB2-treated cells compared to siMock-treated cells. Moreover, these interactions last longer in time. These results are described in the last part of the “Results section” (lines 539-547). Graphs with signal quantification have been included in Figure 7 together with the result of RNAP2 dynamic, measured by FRAP. We also added representative images of this experiment in Figure 7 —figure supplement 1.

In parallel, because the results of these PLA “RNAP2-UV-lesions” revealed that depletion of XAB2 influence (directly or indirectly) the retention of RNAP2 on UV-lesions, we decided to perform a PLA experiment between XAB2 and UV lesions after irradiation. The results obtained show that XAB2 interacts directly with or is in proximity of 6-4PP and CPD lesions until their removal and because this is the first time that such a result is obtained, we tool the initiative to add these results in the manuscript in Figure 2 and figure supplement 2. To describe these results, new text has been implemented in the “Results section”, lines 121-127.

2) Resolve the conflict with Nakatsu et al. 2000 and Kuruoka et al. 2008 regarding the reported increase in transcription with RNAseq experiments with siXAB2 +/- UV. This may resolve the conflict or may not, see #5.Editor list5) Regardless of the outcome of point #2, this conflict will need to be resolved, which may need additional experimentation, if the outcome in inconclusive or cannot explain the discrepancy.

Unfortunately, the RNAseq experiments to resolve the conflict with Nakatsu at al. 2000 and Kuruoka et al. 2008 could not be performed because this technique finally requires too much development for our lab at the moment.

Nevertheless, we could resolve the conflict with Nakatsu at al. 2000 by changing the time of EU incubation. Indeed, with 1 hour of EU incubation (the same time frame used in the work of Nakatsu) the quantity of nascent RNA is reduced in absence of XAB2 compared to control cells. The difference of nascent RNA synthesis between 1h and 2h of EU incubation could be explained by the role of XAB2 in transcription as well as in splicing. New text has been implemented in the “Results section”, lines 102-107.

We decide not to explore further this result because we consider it is not in the main scope of this article and it does not change our conclusions.

3) Benzonase treatment in IP XAB2 experiments represents a critical control.

As suggested by the reviewers #4, we performed Benzonase treatment to confirm the interaction of XAB2 with R-loops. We observed that after benzonase treatment and immunoprecipitation with S9.6 antibody, XAB2 is not anymore revealed by western blot, confirming the interaction of XAB2 with DNA:RNA hybrids. The figure and the text corresponding to this part of result have been changed, lines 390-394 and Figure 5B.

4) The plans for the expanded discussion and the changes already implemented are fine.6) I agree that additional experimentation regarding RNAP1 is not needed in the revision, but the conclusion will have to be significantly toned down and limitations be discussed regarding the claimed RNAP2 specificity.

We changed/added text in the ‘Discussion section’ according to the new results:

Moreover, we tone down our conclusion about the specificity of XAB2 to repair only RNAP2 genes (line 604-607)

7) The authors should tone down the claim that S9.6 is specific for R-loops, the antibody recognizes also other nucleic acid structures. See J.A. Smolka, L.A. Sanz, S.R. Hartono, F. Chedin Recognition of RNA by the S9.6 antibody creates pervasive artifacts when imaging RNA:DNA hybrids J. Cell Biol., 220 (2021).

In the paper of Smolka et al. 2021, it has been demonstrated that S9.6 signal is prominently cytosplasmic and nucleolar and derive mainly from ribosomal RNA. It was not well described in our manuscript in the first version, but when we performed immunofluorescence with S9.6 antibody, we removed the cytoplasm before fixation and; during quantification, the nucleolar signal was subtracted from the nuclear signal. This information was described in Figure 5 —figure supplement 2 and new text has been implemented in the ‘Results section’, line 395-400.

A new paragraph (line 644-654) was added in the ‘Discussion section’ to precise that although a lot of controls have been performed to confirm the specificity of the interaction of XAB2 and AQR with R-loops, we cannot exclude that these proteins release from other nucleic acid structure, for example double-stranded DNA.

[Editors' note: further revisions were suggested prior to acceptance, as described below.]

The manuscript has been improved but there are some remaining issues that need to be addressed, as outlined below:The revision addresses the essential points in the previous review. The manuscript requires extensive language editing. There are numerous problems already in the abstract (line 4 'demonstrate', line 7 'becomes', line 8 'change in mobility is restored' needs rephrasing, line 10 'from DNA:RNA hybrids', line 11 'proteins', line 13 'demonstrate').Reviewer #4 (Recommendations for the authors):The addition of PLA data for the interaction of both RNAP and XAB2 with UV photoproducts is very welcome, and the data quite clearly show increased interaction with damage for both proteins. However, these results actually demand some rethinking of the conclusions.The authors addressed the essential points for the revision but the manuscript requires extensive language editing.

We would like to thank the referees for having reviewed positively our manuscript for ‘*ELife*’. As requested, we tried to make extensive language editing but we are not English speaking. The new uploaded Article File include tracked changes indicating the revisions made, using the tracked changes function in Word. The following texts have been extensively changed:

Results part – XAB2 dynamic during TC-NER – 1^st^ paragraph.

Old version:

“In this technique, fluorescence molecules are photo-bleached in a small spot by a high intensity laser pulse; then the recovery of fluorescence within the bleached area is monitored over time. With no treatment, this measure of fluorescence recovery corresponds to the protein mobility within the living cells. After perturbation of the nuclear environment (e.g. DNA damage), a protein can become less mobile if it physically interacts with a new substrate or a bigger complex; more mobile if the protein is released from its substrate; or eventually will not change its mobility.”

New version:

“In this technique, fluorescence molecules are photo-bleached in a small spot by a high-intensity laser pulse. Subsequently, fluorescence recovery within the bleached area is monitored over time. When cells are untreated, the measure of fluorescence recovery corresponds to the protein intrinsic mobility within the living cells. After perturbation of the nuclear environment (e.g., DNA damage), a protein can physically interacts with a new substrate or a slower complex, becoming less mobile or on the contrary can be released from its substrate, becoming more mobile. Eventually, the protein can also have an unchanged mobility.”

Results part – XAB2 is not released from the splicing complex during DNA repair reactions – 2^nd^ paragraph.

Old version:

“In order to distinguish between these two possibilities, we firstly investigated whether XAB2 dissociates after DNA damage induction from the splicing complex described earlier. XAB2 was immunoprecipitated together with AQR. Interestingly, XAB2 was strongly and consistently immunoprecipitated 1-hour post-irradiation which corresponded to the time in which XAB2 mobility is increased and at the same time point more AQR is also immunoprecipitated. No clear release of XAB2 from AQR was observed at different time points.”

New version:

“In order to distinguish between these two possibilities, we firstly investigated whether, after DNA damage induction, XAB2 dissociates from its splicing partner AQR by immunoprecipitating XAB2 and AQR. Interestingly, XAB2 was immunoprecipitated more strongly and consistently 1-hour post-irradiation, time that corresponds to the XAB2 mobility increase. At the same time point, more AQR is also immunoprecipitated. No clear release of XAB2 from AQR was observed at different time points.”

Results part – XAB2 is released from R-loops during DNA repair reactions – End of this part.

Old version:

“Next, we verify whether XAB2 increase in mobility after UV irradiation is due to a decreased interaction with messenger RNA (mRNA). A decrease in the interaction between XAB2 and mRNA was observed but this decrease corresponded to the decrease of mRNA caused by UV-dependent transcription inhibition invalidating the results observed in the PLA XAB2-mRNAs.”

New version:

“Next, we verified whether the increase in XAB2 mobility observed after UV irradiation is due to a decreased interaction with messenger RNA (mRNA). PLA results show that a reduction in the interaction between XAB2 and mRNA was observed. However, this reduction paralleled the decrease of mRNA caused by UV-dependent transcription inhibition, suggesting that the results observed in the PLA XAB2-mRNAs are due mainly to a decrease in mRNAs amount.”

[Editors' note: this paper was reviewed by Review Commons.]

General Statements

We would like to thank the referees for having reviewed our manuscript. Their comments and requests will help us to improve our manuscript.

Description of the planned revisions

Reviewer 3 thinks that the enhanced mobility of RNA polymerase 2 (RNAP2) after UV due to the absence of XAB2 might be a central observation that should be further explored. As suggest by the reviewer, we will perform PLA with the RNAP2 and CPD or 6-4PP antibody with or without UV and/or siXAB2 to observe if stalling RNAP2 might be affected.

Our figure S2B is in direct contrast with published findings (Nakatsu et al., 2000; Kuraoka et al. 2008) and with our FRAP experiment on RNAP2 as mentioned by reviewer 4 (point 1 and 2) and also reviewer 1. Indeed, we observed by fluorescence of EU incorporation more RNA in siXAB2 treated cells compared to siMock treated cells, without any damage. An increase of RNA is frequently correlated with an increase in transcription. However, as XAB2 is a splicing factor, we could hypothesize that the observed increased RNA may be linked to splicing defect. Indeed, the paper of Goulielmaki et al. 2021 already show by RNAseq analysis a global impairment of splicing machinery in siXAB2 cells. Nonetheless, this assay was performed with total RNA without any damage. We propose to repeat this experiment of RNAseq in siMock and siXAB2 cells on nascent RNA and in cells UV-irradiated or not.

We will repeat the IP XAB2 experiment with benzonase digestion as propose by reviewer 4 (point 5). Benzonase remove nucleic acid and by consequence R-loops. The experiment will probably explain why the amount of immunoprecipitated XAB2 increased after UV.

The discussion will be changed according to the new result and to take into account all the comments of reviewer (for example point 3, 6, 7 and 9 of reviewer 4). A hypothesis will also be proposed. The numerous typographical, grammatical and language errors throughout the manuscript noted by reviewer 4 will be corrected at the end.

Description of the revisions that have already been incorporated in the transferred manuscript

We have already carry out the following minors modifications in the transferred manuscript as suggest by reviewer 2, 3 and 4.

Description of analyses that authors prefer not to carry out

Reviewer 1 suggest to present the quantification results of PLA as foci count rather than signal intensity. This method is not applicable in our PLA because with obtain many foci which are difficult to distinguish. Moreover, there is a direct correlation between the number of foci and the signal intensity.

Reviewer 2 ask to include in Figure 2 images of baseline (NoUV). This request is not necessary as the fluorescence in the local damage is compared to the fluorescence of the whole nucleus without local damages and without nucleolus.

Reviewer 4 (point 4) find our RNAFish 47S not strong enough to conclude that XAB2 is not involved in TCR of RNAP1-transcribed gene because the recovery in WT cells is higher (150%) than the starting value (without irradiation, 100%), whereas in siXAB2 cells it returns exactly to pre-damage level. Usually, in WT cells the recovery is similar to non-irradiated condition but like any living organism the results are never completely reproducible. In conclusion, we will not perform further experiment to strengthen our RNAFIsh 47S result.

Reviewer 2 ask why we used a student’s test while data are presented with 3 or more groups, for which this test is not valid. We will not change our statistic method as we always compared only two different conditions: after irradiation compared to No UV condition or siRNA X compared to siMock. It was not specified in our manuscript and we correct this forgets in the figure legends and in the methods section.

Reviewer 2 and reviewer 4 would like better western blot pictures. We already put in the manuscript our best representative western blot.